# Divergent downstream biosynthetic pathways are supported by L-cysteine synthases of *Mycobacterium tuberculosis*

**Mehak Zahoor Khan**[1,2], **Debbie M Hunt**[3], **Biplab Singha**[1,2], **Yogita Kapoor**[2,4], **Nitesh Kumar Singh**[2], **D V Sai Prasad**[5], **Sriram Dharmarajan**[5], **Divya Tej Sowpati**[2], **Luiz Pedro S de Carvalho**[3,6]*, **Vinay Kumar Nandicoori**[1,2,4]*

[1]National Institute of Immunology, New Delhi, India; [2]CSIR-Centre for Cellular and Molecular Biology, Hyderabad, India; [3]The Francis Crick Institute, London, United Kingdom; [4]Academy of Scientific and Innovative Research (AcSIR), Ghaziabad, India; [5]Department of Pharmacy, Birla Institute of Technology and Science-Pilani, Hyderabad, India; [6]Department of Chemistry, The Herbert Wertheim UF Scripps Institute for Biomedical Innovation & Technology, Jupiter, United States

**\*For correspondence:**
soriodecarval.lp@ufl.edu (LPSdC);
vinaykn@nii.ac.in (VKumarN)

**Competing interest:** The authors declare that no competing interests exist.

**Abstract** *Mycobacterium tuberculosis*'s (*Mtb*) autarkic lifestyle within the host involves rewiring its transcriptional networks to combat host-induced stresses. With the help of RNA sequencing performed under various stress conditions, we identified that genes belonging to *Mtb* sulfur metabolism pathways are significantly upregulated during oxidative stress. Using an integrated approach of microbial genetics, transcriptomics, metabolomics, animal experiments, chemical inhibition, and rescue studies, we investigated the biological role of non-canonical L-cysteine synthases, CysM and CysK2. While transcriptome signatures of *RvΔcysM* and *RvΔcysK2* appear similar under regular growth conditions, we observed unique transcriptional signatures when subjected to oxidative stress. We followed pool size and labelling ($^{34}$S) of key downstream metabolites, viz. mycothiol and ergothioneine, to monitor L-cysteine biosynthesis and utilization. This revealed the significant role of distinct L-cysteine biosynthetic routes on redox stress and homeostasis. CysM and CysK2 independently facilitate *Mtb* survival by alleviating host-induced redox stress, suggesting they are not fully redundant during infection. With the help of genetic mutants and chemical inhibitors, we show that CysM and CysK2 serve as unique, attractive targets for adjunct therapy to combat mycobacterial infection.

## eLife assessment

Sulphur atoms derived from cysteine are thought to play significant roles in maintaining redox homeostasis in Mycobacterium tuberculosis, which encounters stresses associated with immune cell interactions. In this **valuable** manuscript, the authors provide **solid** evidence that the genes encoding cysteine biosynthetic enzymes (cysM and cysK2) are required to maintain full viability of M. tuberculosis under in vitro stress conditions, macrophage infections, and within the lung tissues of mice. The manuscript presents transcriptomic and metabolomic evidence to support the hypothesis that CysM and CysK2 play distinct roles in maintaining cysteine-derived metabolite pools under stress conditions. The work will be of interest to microbiologists in general.

## Introduction

*Mycobacterium tuberculosis* (*Mtb*) continues to stride as the number one killer among all infectious diseases, accounting for nearly 1.5 million deaths yearly. The aggravating situation is despite the clinical use of over 20 antibiotics and a century-old vaccine, BCG. The gradual rise in the emergence

of increasingly drug-resistant strains and HIV-TB (Human Immunodeficiency Virus-Tuberculosis) co-infection further highlights the urgency to identify newer attractive drug targets. Throughout the course of infection, *Mtb* is exposed to a continuum of dynamic host-induced stresses such as severe nutrient deprivation, acidified compartments, and toxic reactive oxygen species (ROS) and reactive nitrogen species (RNS) produced by its resident phagosomes. In turn, *Mtb* produces copious amounts of actinomycetes-specific mycothiol, the major antioxidant in actinomycetes that act as the functional equivalent of glutathione, to combat ROS and RNS. In addition to mycothiol, *Mtb* also produces ergothioneine, a low molecular weight thiol, and several enzymes that act concertedly to subvert host-induced redox stress. The redox-active group of both mycothiol and ergothioneine is derived from L-cysteine. Hence, genes involved in the biosynthesis of L-cysteine are upregulated in the host and in vitro upon oxidative and nutritional stress (*Hampshire et al., 2004*; *Manganelli et al., 2002*; *Pinto et al., 2004*; *Schnappinger et al., 2003*; *Khan et al., 2021*). Notably, an increased expression of these genes is functionally crucial, as suggested by the attenuated survival of transposon mutants of many sulfur and L-cysteine biosynthesis genes within the host (*Rengarajan et al., 2005*). In mycobacteria, sulfur assimilation begins with the import of sulfate through a sulfate transporter composed of SufI.CysT.W.A. Intracellular sulfate is a substrate for APS synthase CysD.N.C, which adenylates and phosphorylates sulfate to form adenosine 5'-phosphosulfate (APS) (*Wooff et al., 2002*; *McAdam et al., 1995*; *Urbanek et al., 1990*). APS sits at a metabolic branch point; it can either be converted into sulfolipids (*Mougous et al., 2002*) by consequent actions of multiple Stfs enzymes or reduced via SirA and SirH to sulfide (*Figure 1—figure supplement 1*). This pathway encompasses sulfide formation from sulfate, called the sulfur assimilation pathway (*Schnell et al., 2005*; *Pinto et al., 2007*) *Mtb* genome encodes three L-cysteine synthases – the canonical CysK1 and non-canonical CysM and CysK2 enzymes. Interestingly, humans do not possess L-cysteine synthases, raising the possibility of developing antibiotics without a homologous target in the host. CysK1 utilizes sulfide produced via the sulfur assimilation pathway and *O*-acetyl-L-serine produced from glycolytic intermediate 3-phosphoglycerate to produce L-cysteine (*Qiu et al., 2013*; *Schnell et al., 2007*; *Agren et al., 2008*; *O'Leary et al., 2008*; *Jurgenson et al., 2008*; *Burns et al., 2005*) CysM, on the other hand, uses *O*-phospho-L-serine and a small sulfur carrier protein CysO as substrates (*Agren et al., 2008*; *Burns et al., 2005*). Like CysK1, CysK2 utilizes *O*-phospho-L-serine and sulfide as substrates (*Steiner et al., 2014*; *Nakamura et al., 1984*; *Figure 1—figure supplement 1*). In addition, *Mtb* can also synthesize L-cysteine through a reverse transsulfuration pathway from L-methionine. This example of convergent metabolic redundancy raises several interesting questions: (1) Why would *Mtb* rely on multiple enzymes and pathways to produce the same biomolecule? (2) Are these 'functionally redundant' enzymes dispensable, or are they required at a distinct cellular space, time, and condition? (3) Is the L-cysteine pool produced through a particular pathway functionally compartmentalized? That is, is it metabolized into a specific kind of downstream thiol?

To define these unsolved aspects of *Mtb* L-cysteine metabolism, we sought to investigate the interplay of non-canonical L-cysteine synthases of *Mtb* and elucidate their roles in abetting virulence. We aimed to decipher the relative contribution of CysM and CysK2 enzymes in alleviating host-induced stresses and promoting the survival of *Mtb* within the host. We also investigated their role in secondary metabolism, synthesizing low molecular weight thiols, such as mycothiol and ergothioneine, and understanding the consequential effects of their deletion on the global transcriptome of *Mtb*. Lastly, with the help of specific inhibitors, we evaluated their potential to serve as attractive drug targets for adjunct antibiotic therapy.

## Results

### Non-canonical L-cysteine synthases facilitate Mtb in combating host-induced stresses

*Mtb* is a generalist, a prototroph organism that can produce all 20 proteinogenic amino acids. In agreement with this notion, numerous microarray studies depict the upregulation of multiple amino acid pathways within the host (*Pinto et al., 2004*; *Schnappinger et al., 2003*; *Pinto et al., 2007*), indicating a higher dependency of *Mtb* survival and virulence on amino acid biosynthesis and regulation. In response to infection, host immune responses often try to contain the bacillary growth by depriving the amino acid levels in intracellular environment. As a counter mechanism, *Mtb* has been shown to

upregulate biosynthesis of amino acids such as tryptophan, lysine, and histidine to facilitate mycobacterial survival within the host (*Zhang et al., 2013*; *Dwivedy et al., 2021*). We sought to identify distinct host stresses that result in the transcriptional modulation of specific amino acid biosynthetic pathways with the help of RNA sequencing (RNA-seq). We compared the transcriptional profile of *H37Rv* (*Rv*) grown in 7H9-ADC with the profiles obtained when bacilli were subjected to oxidative, nitrosative, starvation, and acidic stresses (*Supplementary file 1*). The volcano plot illustrates differentially expressed genes (DEGs) that were significantly upregulated (blue) and downregulated (red) under indicated stress conditions (absolute $\log_2$ Fold change >1 and $p_{adj}$ <0.05) (*Figure 1—figure supplement 2a–e*). Exposure to starvation conditions resulted in drastic transcription modulation compared with other stresses, suggesting that nutrient deficiency is the primary driver of transcriptome remodelling. In contrast, we observed the lowest number of DEGs when the bacteria were subjected to mildly acidic conditions (pH 5.5). Heat maps of normalized DEGs depict that DEG changes were comparable across biological replicates within each sample set (*Figure 1—figure supplement 2f–j*). To better understand the RNA-seq results, we plotted the fold change of DEGs due to different stress conditions (*Figure 1—figure supplement 3* and *Supplementary file 2*). This allowed us to understand the expression profile of genes in all the stress conditions simultaneously, regardless of whether they were identified as differentially expressed. It is apparent from the data that a specific cluster of genes is up- and downregulated in oxidative, sodium dodecyl sulfate (SDS), and starvation conditions. In comparison, the differences observed in the pH 5.5 and nitrosative conditions were limited (*Figure 1—figure supplement 3* and *Supplementary file 2*).

To further refine our understanding of the DEGs, we grouped them into various functional categories and found that genes belonging to intermediary metabolism and respiration remained the most affected in all conditions highlighting their role of metabolic rewiring (*Figure 1—figure supplement 4a*). Pathway enrichment analysis of the most enriched Gene Ontology (GO) further revealed that, while, expectedly, metabolic pathways were found to be downregulated during starvation, we observed enrichment of nitrogen metabolism (*Figure 1—figure supplement 4b* and *Supplementary file 3*). SDS stress resulted in the upregulation of branched-chain amino acids/keto acids degradation pathways (*Figure 1—figure supplement 4c*), and nitrosative stress ensued upregulation of fatty acid and lipid biosynthetic processes (*Figure 1—figure supplement 4d*). There were no changes observed in mildly acidic conditions (*Figure 1—figure supplement 4e*). Importantly, oxidative stress resulted in significant upregulation of genes involved in sulfur metabolism (*Figure 1—figure supplement 4f*). We further analyzed the DEGs involved in sulfur and L-cysteine metabolism across sample sets and discerned an overlap of the genes affected under two or more conditions. Interestingly, we observed upregulation of sulfate transporters genes (*subI*, *cysT*, *cysW*, and *cysA1*) across multiple stresses and sulfur assimilation (*cysH*, *cysA2*, *cysD*, and *cysNC*) during starvation, oxidative and SDS stress. *CysK2*, a non-canonical L-cysteine synthase, was found to be upregulated during all stresses except SDS stress (*Figure 1a*).

Thus RNA-seq data suggest that genes involved in sulfur assimilation and L-cysteine biosynthetic pathway are upregulated during various host-like stresses in *Mtb* (*Figure 1—figure supplement 4*). Given the importance of sulphur metabolism genes in in vivo survival of *Mtb* (*Hatzios and Bertozzi, 2011*; *Senaratne et al., 2006*) it is not surprising that diverse environment cues dynamically regulate these genes. Microarray studies have shown upregulation of genes encoding sulphate transporter upon exposure to hydrogen peroxide and nutrient starvation (*Hampshire et al., 2004*; *Schnappinger et al., 2003*; *Betts et al., 2002*; *Voskuil et al., 2011*; *Voskuil et al., 2004*). Similarly, ATP sulfurylase and APS kinase are induced during macrophage infection and by nutrient depletion. Induction of these genes that coordinate the first few steps of the sulfur assimilation pathway indicates a probable increase in biosynthesis of sulfate-containing metabolites that may be crucial against host-inflicted stresses. Furthermore, genes involved in synthesis of reduced sulfur moieties (*cysH*, *sirA*, and *cysM*) are also induced by hydrogen peroxide and nutrient starvation. Sulfur metabolism has been postulated to be important in transition to latency. This hypothesis is based on transcriptional upregulation of *cysD*, *cysNC*, *cysK2*, and *cysM* upon exposure to hypoxia. Multiple transcriptional profiling studies have reported upregulation of *moeZ*, *mec*, *cysO*, and *cysM* genes when cells were subjected to oxidative and hypoxic stress (*Manganelli et al., 2002*; *Burns et al., 2005*; *Hatzios and Bertozzi, 2011*; *Voskuil et al., 2011*; *Voskuil et al., 2004*; *Brunner et al., 2017*; *Manganelli et al., 2001*) further suggesting an increase in the biosynthesis of reduced metabolites such as cysteine and

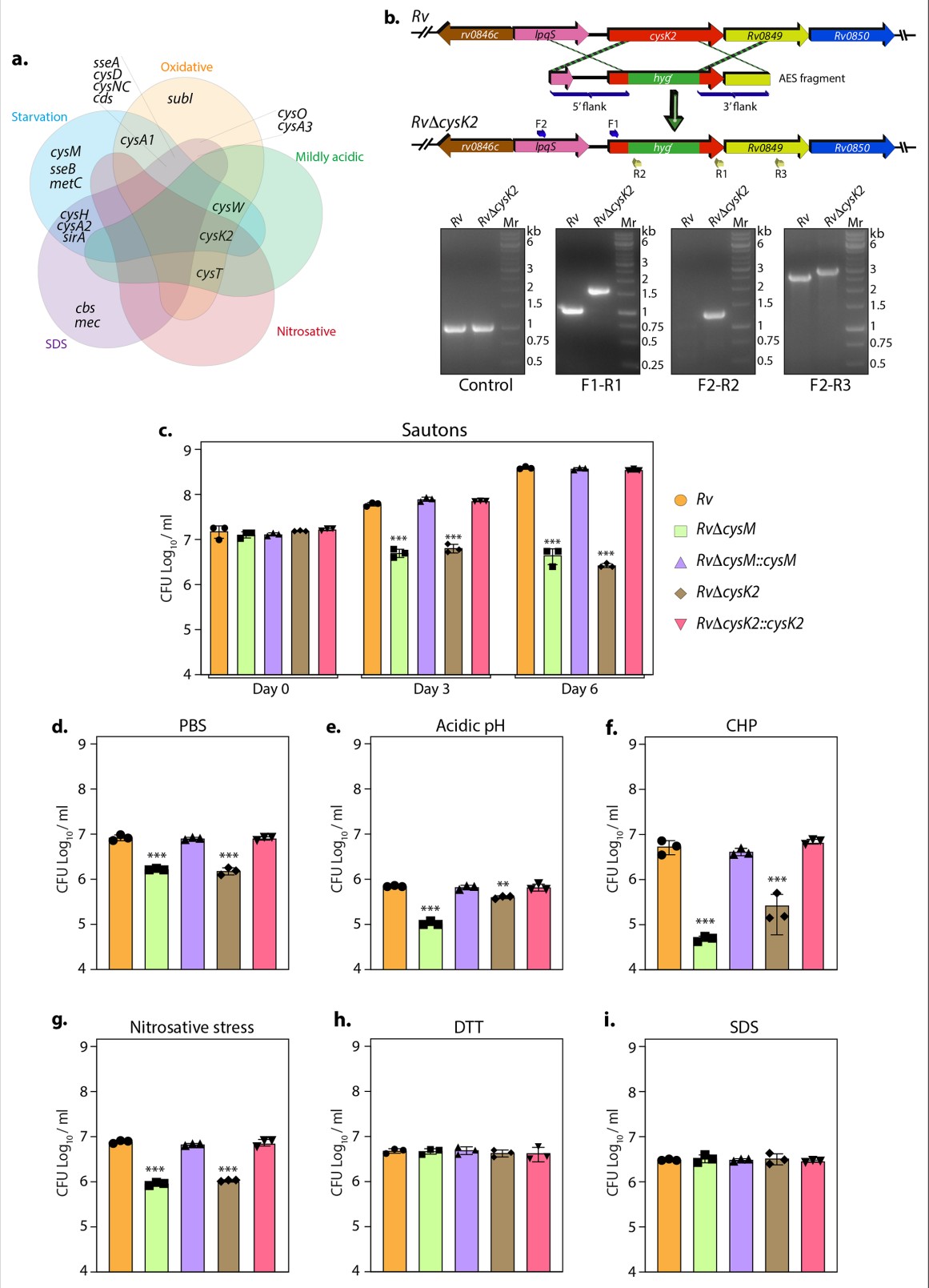

**Figure 1.** Non-canonical L-cysteine synthases facilitate mycobacterial survival upon host-like induced stresses in vitro. (**a**) Five-way Venn diagram highlighting differentially expressed genes (DEGs) belonging to sulfur metabolism pathway under indicated stress conditions (absolute log$_2$ Fold change >0.5 and p$_{adj}$ value <0.05). (**b**) Line diagram illustrating the *cysK2* loci in *Rv* and *RvΔcysK2* and the strategy employed for the replacement of *cysK2* with *hyg$^r$*. Primer sets used for confirming the generation of *RvΔcysK2* are indicated as arrows. The first agarose gel image shows PCR (Polymerase

*Figure 1 continued on next page*

*Figure 1 continued*

chain reaction) amplicons with a control (*sigB*) gene-specific primer indicating the presence of nearly equivalent amount of genomic DNA isolated from *Rv* and *RvΔcysK2*. The second panel shows PCR amplicons with gene-specific primer set (F1–R1) in *Rv* and *RvΔcysK2* mutant, the third panel shows amplicons (F2–R2) expected only in *RvΔcysK2* mutant, and the fourth panel shows amplicons (F2–R3) in *Rv* and *RvΔcysK2* mutant confirming legitimate recombination at native loci. Mr represents 1 kb gene ruler ladder. *Rv*, *RvΔcysM*, *RvΔcysK2*, *RvΔcysM::cysM*, and *RvΔcysK2::cysK2* strains were inoculated in Sauton's (**c**), PBS (Phosphate-Buffered Saline) (**d**), or acidic (**e**), oxidative 50 µM cumene hydroperoxide (CHP) for 24 hr (**f**), nitrosative (**g**), reductive (**h**), and sodium dodecyl sulfate (SDS) (**i**). Bar graphs represent the bacillary survival with data point indicating values CFU $\log_{10}$/ml ± standard deviations (SDs) from individual replicate ($n = 3$). Statistical significance was drawn in comparison with *Rv* using one-way analysis of variance (ANOVA) followed by a post hoc test (Tukey test; GraphPad prism). **$p < 0.005$; ***$p < 0.0005$.

The online version of this article includes the following source data and figure supplement(s) for figure 1:

**Source data 1.** Raw CFU/ml values for *Figure 1c–i*.

**Source data 2.** Raw unedited gel images for *Figure 1b*.

**Source data 3.** Uncropped and labelled images for *Figure 1b*.

**Figure supplement 1.** Overview of sulfur metabolism pathway in *Mycobacterium tuberculosis*.

**Figure supplement 2.** Sulfur assimilation and L-cysteine biosynthesis pathways are upregulated under stress conditions.

**Figure supplement 3.** Genes that were differentially expressed in at least one of the five stresses.

**Figure supplement 4.** Genes belonging to the sulfur metabolism pathway are upregulated upon oxidative stress.

**Figure supplement 5.** L-Cysteine synthases facilitate mycobacterial survival upon oxidative stresses in vitro.

**Figure supplement 5—source data 1.** Raw CFU/ml values for *Figure 1—figure supplement 5a–c*.

methionine and sulfur containing cell wall glycolipids upon exposure to oxidative stress. To address the functional relevance of this observation, we deleted two non-canonical L-cysteine synthases-CysM (*Khan et al., 2021*) and CysK2, from the *Mtb* chromosome. CysK2 mutant, *RvΔcysK2*, was generated using the recombineering method, and the recombination at the native loci was confirmed with the help of multiple PCRs (*Figure 1b*). While deletion of *cysM* or *cysK2* did not affect mycobacterial growth under in vitro nutrient-rich 7H9-ADC or 7H9-ADS, it significantly compromised *Mtb* growth in defined Sauton's media (~2 $\log_{10}$; *Figure 1c*), suggesting the importance of *cysM*- and *cysK2*-derived L-cysteine. Restoring *cysM* or *cysK2* expression in the complementation strains, *RvΔcysM::M* and *RvΔcysK2::K2*, rescued the growth defects (*Figure 1—figure supplement 5* and *Figure 1c*). *Mtb* is a metabolically versatile organism capable of utilizing a large variety of carbon and nitrogen sources (*de Carvalho et al., 2010*). Unlike 7H9, wherein glucose, glycerol, glutamate, and ammonia act as carbon and nitrogen sources, Sauton's media contains glycerol and asparagine as the sole carbon and nitrogen sources, suggesting metabolic reprogramming aided by CysM and CysK2 enable *Mtb* to grow optimally in a limited nutritional environment. This observation was further recapitulated under nutrient starvation (PBS), wherein the survival of *RvΔcysM* and *RvΔcysK2* was lower than parental *Rv* or *RvΔcysM::M* or *RvΔcysK2::K2* strains (*Figure 1d*). Interestingly, when exposed to acidic conditions (pH 4.5), *RvΔcysM* and *RvΔcysK2* survival were observed to be ~0.85 and ~0.24 $\log_{10}$ lower, respectively, compared with *Rv*, suggesting CysM is relatively more important for bacillary survival under acidic stress (*Figure 1e*). The highest attenuation of *RvΔcysM* and *RvΔcysK2* was observed upon the addition of cumene hydroperoxide (CHP), an organic hydroperoxide that, upon decomposition, generates free radicals. The relative survival of *RvΔcysM* and *RvΔcysK2* was ~2.04 and ~1.31 $\log_{10}$ lower, respectively, compared to *Rv* (*Figure 1f*). The addition of diamide, which results in thiol oxidation, also attenuated the survival of *RvΔcysM* and *RvΔcysK2* by ~1.33 and ~1.26 $\log_{10}$ compared with *Rv* (*Figure 1—figure supplement 5c*). When the mutant strains were subjected to nitrosative stress, the survival of *RvΔcysM* and *RvΔcysK2* was ~0.93 and ~0.86 $\log_{10}$ lower than *Rv* (*Figure 1g*). However, no significant attenuation was found during reductive and SDS stress (*Figure 1h, i*). Collectively, *RvΔcysM* and *RvΔcysK2* displayed increased susceptibility towards oxidative, nitrosative, mild acidification, and PBS starvation to varying degrees compared with *Rv*, *RvΔcysM::M*, and *RvΔcysK2::K2*. The data suggest that L-cysteine, produced via CysM and CysK2, and its downstream products help mycobacteria thwart specific stresses that *Mtb* encounters within the host.

## Distinct roles of CysM and CysK2

To decipher the mechanism through which CysM and CysK2 combat oxidative stress, we performed a global transcriptomic analysis of *Rv*, *RvΔcysM*, and *RvΔcysK2* in the presence and absence of oxidative

stress (CHP) (*Supplementary file 4*). Principal component analysis demonstrated clear separation of strains under different conditions. Intriguingly, while *RvΔcysM* and *RvΔcysK2* were closely located on a PCA (Principal Component Analysis) plot in the absence of any stress, oxidative stress resulted in a significant divergence between the two groups (*Figure 2a*). Deletion of *cysM* resulted in differential expression of 322 genes (159 downregulated and 163 upregulated) (*Figure 2—figure supplement 1a, b*), while deletion of *cysK2* impacted 278 genes (155 downregulated and 123 upregulated) under regular growth conditions (*Figure 2—figure supplement 1c, d*) (absolute log$_2$ Fold change >1 and p$_{adj}$ <0.05). In contrast, upon treatment with CHP, nearly ~33% and ~53% of *Mtb* genes were differentially expressed in *RvΔcysM* and *RvΔcysK2*, respectively, compared with *Rv* (*Figure 2b, c*). To understand the individual contribution of CysM and CysK2 in combating oxidative stress, we compared the DEGs of *RvΔcysK2* to *RvΔcysM* under regular growth and oxidative stress conditions. While DEGs between *RvΔcysK2* and *RvΔcysM* were limited to 13 in untreated conditions (*Figure 2d*), CHP treatment resulted in differential expression of 1372 genes (*Figure 2e* and *Figure 2—figure supplement 1e*), highlighting unique transcriptional signatures associated with each L-cysteine synthase under oxidative stress. Subsequently, we compared the DEGs of *Rv*, *RvΔcysM*, and *RvΔcysK2* obtained under oxidative stress with the help of a Venn diagram. This analysis revealed that CysM and CysK2 influence a shared repertoire of 1023 genes (529 upregulated and 494 downregulated) compared to *Rv* upon CHP treatment (*Figure 2f*). Importantly, CysM and CysK2 distinctively modulate 324 and 1104 genes, respectively, during oxidative stress (*Figure 2f, g*). In addition to being directly regulated by the synthases, it is highly that the changes in the expression of these genes is because of distinct downstream consequences. Assortment of DEGs into functional classes revealed intermediary metabolism and respiration and cell wall pathways among the most impacted categories (*Figure 2h*). These results corroborate the recent finding that CysK2 alters the phospholipid profile of the *Mtb* cell envelope (*Sao Emani et al., 2022*). Pathway enrichment analysis of the most enriched GO biological process indicated that while most pathways are commonly affected by CysM and CysK2, these L-cysteine synthases also have a unique transcriptional footprint (*Supplementary file 5*). Pathways such as DNA binding, homologous recombination, translation, and ribosome were commonly affected in *RvΔcysM* and *RvΔcysK2* compared to *Rv* during oxidative stress (*Figure 2—figure supplement 1f, g*). Genes belonging to oxidative phosphorylation and quinone binding were upregulated, while DNA binding and cholesterol catabolism gene categories were found to be downregulated in *RvΔcysM* compared to *RvΔcysK2* during oxidative conditions (*Figure 2i*). Together, our data point to the unique roles of CysM and CysK2, as their deletion differentially affects various cellular pathways under oxidative stress.

## Key metabolites are differentially affected upon CysM and CysK2 deletion

L-Cysteine concentration in *Mtb* is notoriously low, usually tenfold lower than most other amino acids (*Agapova et al., 2019*). As expected, L-cysteine levels were below the limit of detection/quantification (not shown). Therefore, we used key downstream metabolites to monitor L-cysteine biosynthesis and utilization. We followed pool size and labelling (Na$^{34}$SO$_4$) of L-methionine, mycothiol/mycothione, and ergothioneine, made from L-cysteine, via three distinct and dedicated metabolic pathways (*Figure 3—figure supplement 1*). First, L-methionine, mycothiol, and ergothioneine concentrations vary from zero to 24 hr, indicating that their pool sizes are not as stable as those of other core metabolites. This is consistent with their function, susceptibility to redox homeostasis, and L-methionine's role in protein synthesis. Second, upon challenge with CHP for 24 hr, L-methionine levels decrease in *Rv*, remain constant in *RvDcysK2*, and increase in *RvDcysM* (*Figure 3a*). Ergothioneine levels are identical in the three strains, and upon treatment with CHP, they are decreased in the parent strain but remain unchanged in the *RvDcysK2* and *RvDcysM* mutant (*Figure 3b*). Mycothiol levels increase in the *Rv*, *RvDcysK2* strains and remain nearly constant in *RvDcysM* (*Figure 3c*). Finally, mycothione levels increased slightly in the parent strain, decreased in the *RvDcysK2*, and remained stable in the *RvDcysM* strain (*Figure 3d*). These changes revealed a significant role of distinct L-cysteine biosynthetic routes on redox stress and homeostasis.

Employing $^{34}$S labelling, we observed a reduced rate of synthesis of mycothiol, mycothione, and ergothioneine in *RvΔcysM* compared to *Rv* and *RvΔcysK2* (*Figure 3e–g*). Interestingly, ergothioneine levels were found to be reduced in both *RvΔcysM* and *RvΔcysK2* compared to *Rv*, possibly underlying one of the reasons for their enhanced sensitivity to oxidative stress.

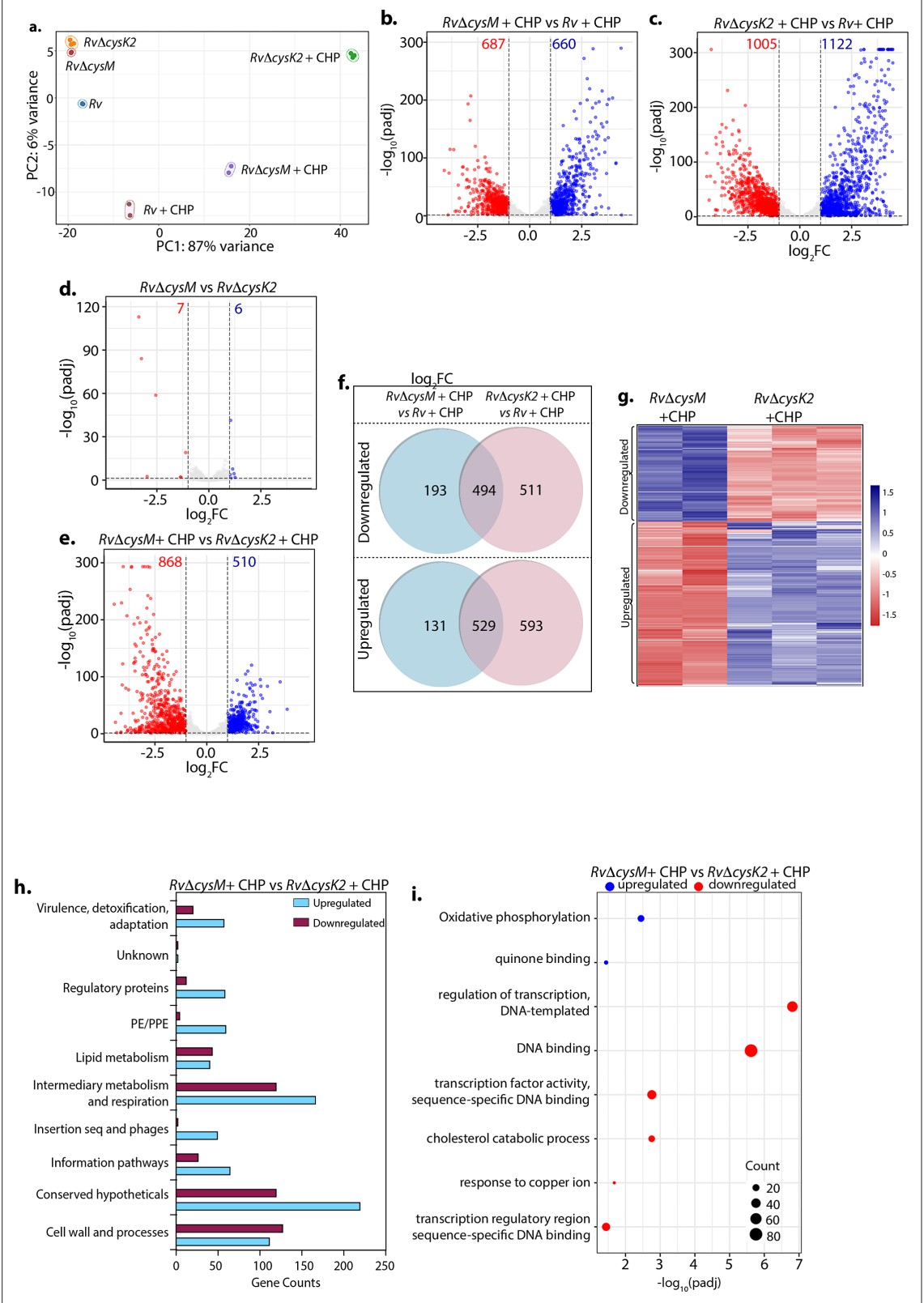

**Figure 2.** Distinct roles of CysM and CysK2 in attuning cellular processes. (**a**) PCA plot demonstrating separation of various bacterial strains under different conditions. (**b–e**) Volcano plots illustrating significantly upregulated (blue) and downregulated (red) genes in indicated strains and conditions with absolute $\log_2$ Fold change >1 and $p_{adj}$ value <0.05. Numbers in the top quadrant highlight the number of significantly upregulated (blue) and downregulated (red) genes in each condition. (**f**) Venn diagram showing the number of significantly down- and upregulated differentially expressed

Figure 2 continued

genes (DEGs) that overlap between indicated strains. (**g**) Heat maps depicting normalized gene count of DEGs in independent replicates of *RvΔcysM* and *RvΔcysK2* grown under oxidative conditions with absolute log$_2$ Fold change >1 and p$_{adj}$ value <0.05. The colour intensity indicates relative upregulated (blue) and downregulated (red) genes compared to the control. (**h**) Horizontal bar graph depicting the number of DEGs belonging to a particular functional category upon oxidative stress in *RvΔcysM* compared to *RvΔcysK2* under oxidative conditions. (**i**) Pathway enrichment by DAVID depicting significantly enriched Gene Ontology (GO) biological processes based on DEGs upon oxidative stress in *RvΔcysM* compared to *RvΔcysK2* (absolute log$_2$ Fold change >1 and p$_{adj}$ <0.05).

The online version of this article includes the following figure supplement(s) for figure 2:

**Figure supplement 1.** Unique transcriptional signatures of CysM and CysK2.

This result further indicates that these two routes are partially redundant; that is, they generate the same end product, L-cysteine. Yet, pool size measurements demonstrate significant changes, highlighting that while these pathways produce the same metabolite, their complex biological roles and requirements (co-substrates, pathways, regulation, etc.) are not fully redundant. This interpretation is in accordance with transcriptomics and phenotypic results described above. These non-overlapping metabolic requirements (e.g., *O*-acetyl-L-serine vs. *O*-phospho-L-serine vs. CysM) are likely the source of the different metabolic phenotypes observed. Therefore, even producing the same end-product, genetic disruption of different L-cysteine synthases differently affects bacterium metabolism and fitness, leading to the distinct phenotypes observed between *RvDcysK2* and *RvDcysM* in vitro, in cellulo, and in vivo.

## CysM and CysK2 alleviate the toxicity of host-produced ROS and RNS

To define whether the attenuation of mutant strains observed during defined in vitro host-like conditions can be recapitulated within the host cells, we compared the survival of parental, mutants, and complementation strains in murine peritoneal macrophages (*Figure 4a*). Survival of *RvΔcysM* and *RvΔcysK2* was ~1.22 and ~0.85 log$_{10}$ lower, respectively, compared with *Rv* at 96 hr post-infection (p.i) (*Figure 4b*). Notably, the addition of L-cysteine alleviated survival differences (*Figure 4c*), suggesting that reduced L-cysteine levels in mutant strains are responsible for their attenuated survival. To further validate that CysM and CysK2 support *Mtb* survival by detoxifying peroxides generated by the host, we compared the survival of strains in the peritoneal macrophages extracted from wild-type C57BL/6 (WT) mice that are proficient in eliciting oxidative and nitrosative stress or murine strains that are deficient in their ability to produce hydrogen peroxide (H$_2$O$_2$) (phox$^{-/-}$) or both oxidative and nitrosative stress (IFNγ$^{-/-}$). As anticipated, regardless of the mice genotype, there was no difference in the survival of *Rv* and complementation strains (*Figure 4d, e*). On the other hand, attenuation of *RvΔcysM* and *RvΔcysK2* observed in WT macrophages is partially reversed in the macrophages obtained from phox$^{-/-}$ mice (*Figure 4d*) and completely nullified in macrophages isolated from IFNγ$^{-/-}$ mice (*Figure 4e*). To further dissect the relative contribution of CysM and CysK2 in combating ROS and RNS independently or collectively, the survival of *Mtb* strains was examined in peritoneal macrophages isolated from WT and phox$^{-/-}$ in the presence or absence of iNOS (Inducible nitric oxide synthase) inhibitor. While the inhibition of ROS or RNS independently only partially alleviated the attenuated survival *RvΔcysM* and *RvΔcysK2*, inhibition of both ROS and RNS by treating peritoneal macrophages isolated from phox$^{-/-}$ with iNOS inhibitor completely nullified survival defects of the mutants (*Figure 4f*). These results suggest that non-canonical L-cysteine synthases, CysM and CysK2, play an essential role in combating host-induced redox stress.

## CysM- and CysK2-derived L-cysteine support mycobacterial survival in vivo

Given the importance of non-canonical L-cysteine synthases in mitigating host-induced redox stress, we next sought to examine whether CysM and CysK2 independently facilitate mycobacterial survival in a murine infection model. We followed disease progression by enumerating colony-forming units (CFUs) of the lung and spleen of the infected mice at day 1, 4, and 8 weeks p.i (*Figure 5a, d*). CFUs enumerated on day 1 showed equal bacillary deposition across strains in the lungs (*Figure 5b, e*). Compared with *Rv*, survival of *RvΔcysM* was ~1.25 log$_{10}$ lower at 4 weeks p.i, which further attenuated to ~1.71 log$_{10}$ at 8 weeks p.i in the lungs (*Figure 5b*). Similarly, dissemination of *RvΔcysM* in the spleen was ~1.63 log$_{10}$ lower at 4 weeks p.i and ~1.72 log$_{10}$ lower at 8 weeks p.i compared with *Rv*

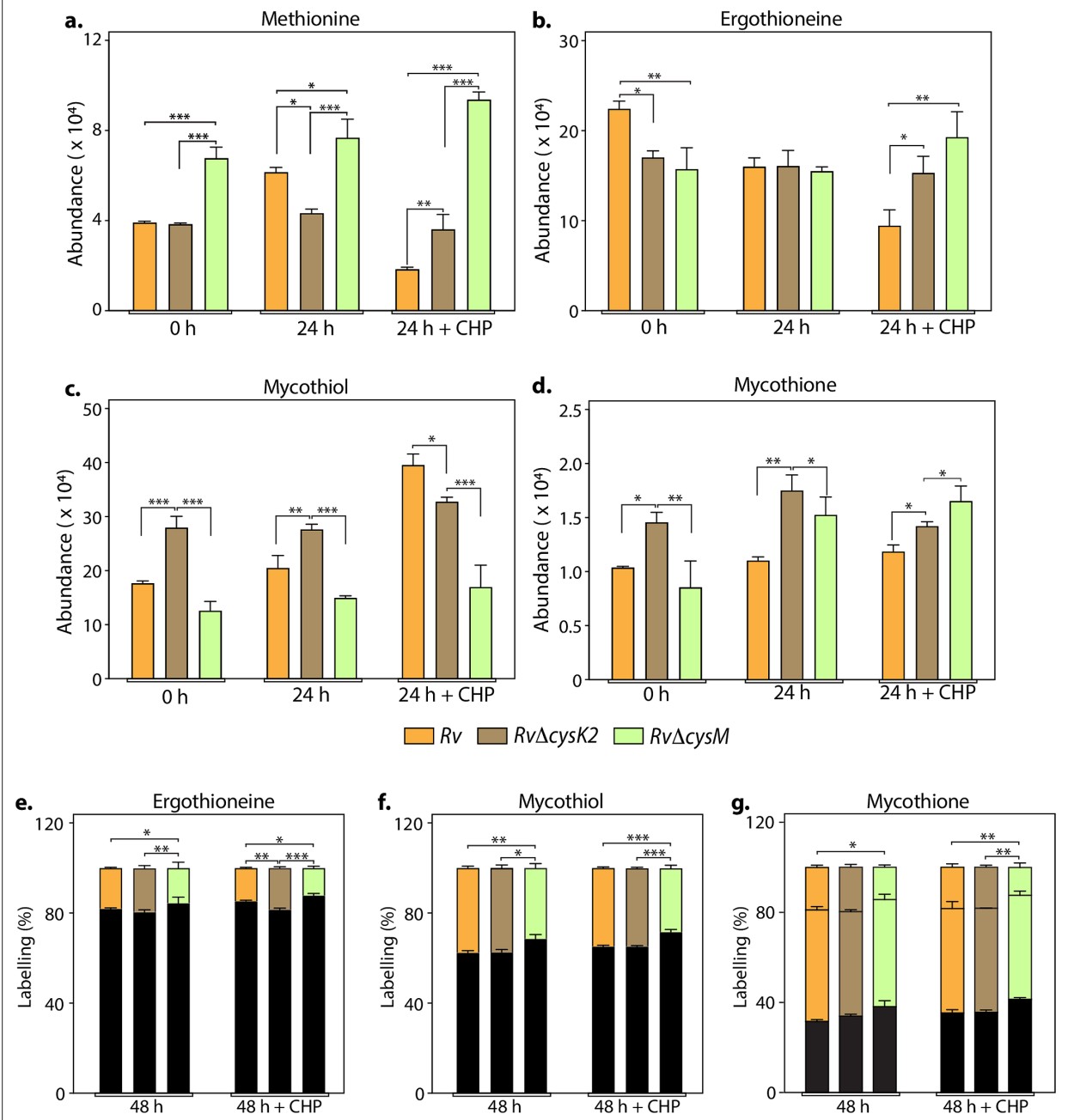

**Figure 3.** Key metabolites are differentially affected upon CysM and CysK2 deletion. Total abundance (ion count/protein concentration) of L-methionine (**a**), ergothioneine (**b**), mycothiol (**c**), and mycothione (**d**) in *Rv strains* with or without 50 µM cumene hydroperoxide (CHP) treatment. The percentage of labelled (coloured) and unlabeled (black) ergothioneine (**e**), mycothiol (**f**), and mycothione (**g**) in *Rv strains* with or without 50 µM CHP treatment at 48 hr. Since mycothione (**g**) has two ionizable sulfur, the two-coloured stacked bar graphs represent percentage of labelled M+2 and M+4 ions. Percentage was calculated with respect to 0 hr abundance for each replicate. Bars depict the mean of biological replicates ($n = 3$), and error bars represent the standard deviation. The same samples were used for analysis (**a–d**) and (**e–g**). Statistical significance was calculated between *Rv* and *RvΔcysM/ RvΔcysK2*; and between *RvΔcysM* and *RvΔcysK2* using non-parametric *t*-test. *$p < 0.05$; **$p < 0.005$; ***$p < 0.0005$.

The online version of this article includes the following source data and figure supplement(s) for figure 3:

**Source data 1.** Raw data values for *Figure 3*.

**Figure supplement 1.** Overview of metabolites tracked through [35]S sulfur incorporation.

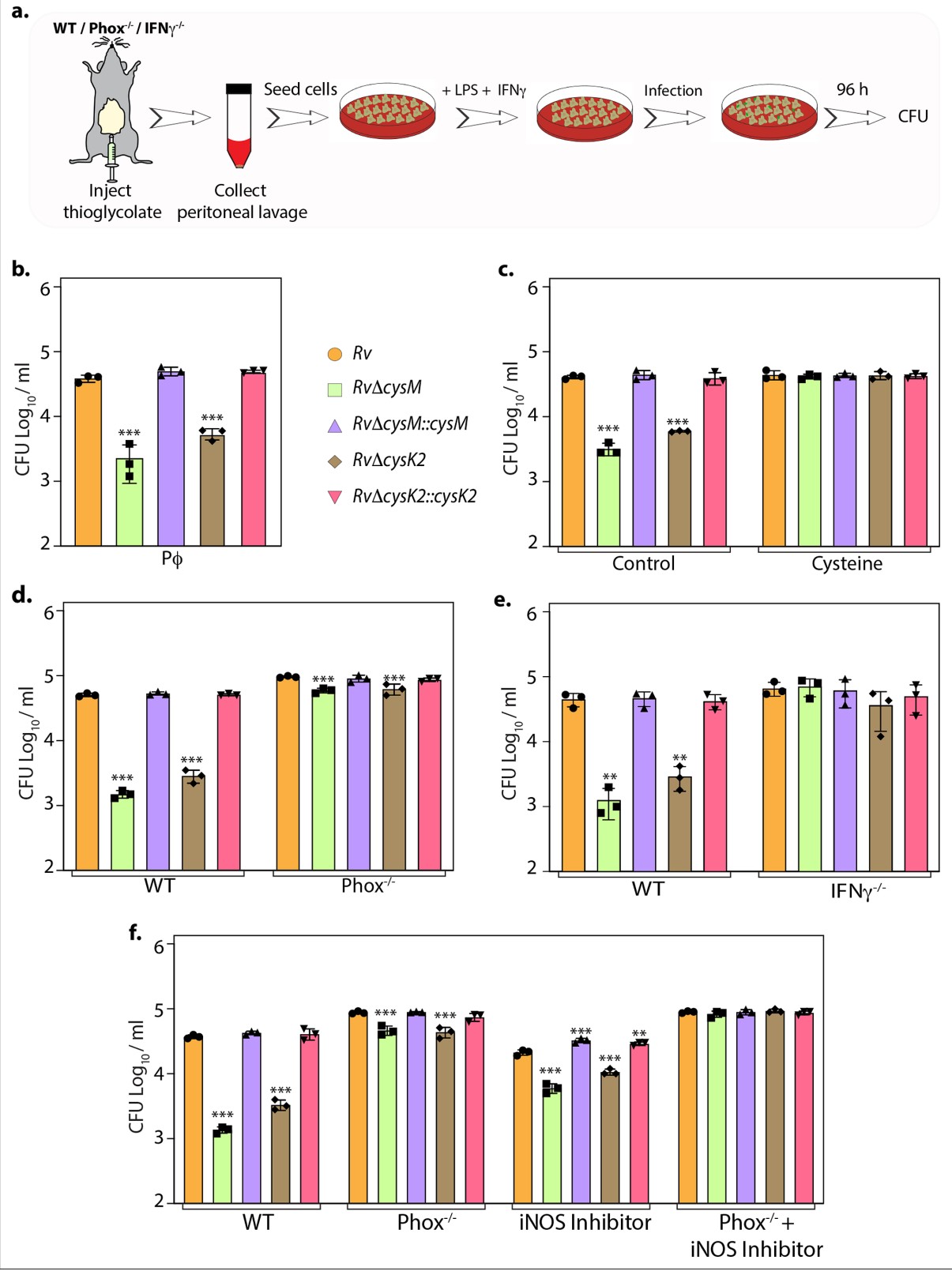

**Figure 4.** L-Cysteine synthases ameliorate mycobacterial survival in response to host-induced oxidative and nitrosative stress. (**a**) Pictorial representation of peritoneal macrophage infection experiments. Thioglycolate was injected into the peritoneum cavity of C57BL/6, phox[−/−], IFNγ[−/−] mice. Four days post-injection, peritoneal macrophages were extracted and activated by treatment with IFNγ (Interferon gamma) overnight and lipopolysaccharide (LPS) 2 hr prior to the infection. In specific cases, iNOS inhibitor 1400 W was added along with IFNγ/LPS. The intracellular bacillary survival was calculated

*Figure 4 continued on next page*

*Figure 4 continued*

96 hr post-infection (p.i.). (**b**) Peritoneal macrophages from C57BL/6 mice were infected, and intracellular bacillary survival was enumerated 96 hr p.i. (**c**) Peritoneal macrophages from C57BL/6 mice were infected, left untreated (control) or treated with 0.2 mM L-cysteine at the time of infection and intracellular bacillary survival was assessed 96 h p.i. (**d–f**) Peritoneal macrophages isolated from mice of indicated genotypes were either left untreated or treated with iNOS inhibitor, 1400 W. Intracellular bacillary survival was enumerated 96 h p.i. (**b–f**) Data points are presented as CFU $\log_{10}$/ml ± standard deviation (SD) of each replicate ($n = 3$). Statistical significance was drawn in comparison with *Rv* using one-way analysis of variance (ANOVA) followed by a post hoc test (Tukey test; GraphPad prism). **$p < 0.005$; ***$p < 0.0005$.

The online version of this article includes the following source data for figure 4:

**Source data 1.** CFU/ml values for *Figure 4*.

(*Figure 5c*). Similarly, *RvΔcysK2* was ~1.67 $\log_{10}$ and ~1.87 $\log_{10}$ lower at 4 and 8 weeks p.i, respectively, compared with *Rv* in the lungs (*Figure 5e*) and ~1.63 $\log_{10}$ and ~1.72 $\log_{10}$ lower at 4 and 8 weeks p.i, respectively, in the spleens of infected mice (*Figure 5f*).

To understand whether the ability of CysM to mitigate oxidative and nitrosative stress is linked to its role in facilitating mycobacterial survival in vivo, we infected WT, phox$^{-/-}$, and IFNγ$^{-/-}$ mice with *Rv*, *RvΔcysM* and *RvΔcysM::M* and analyzed bacillary load at 4 weeks p.i (*Figure 5g*). As shown

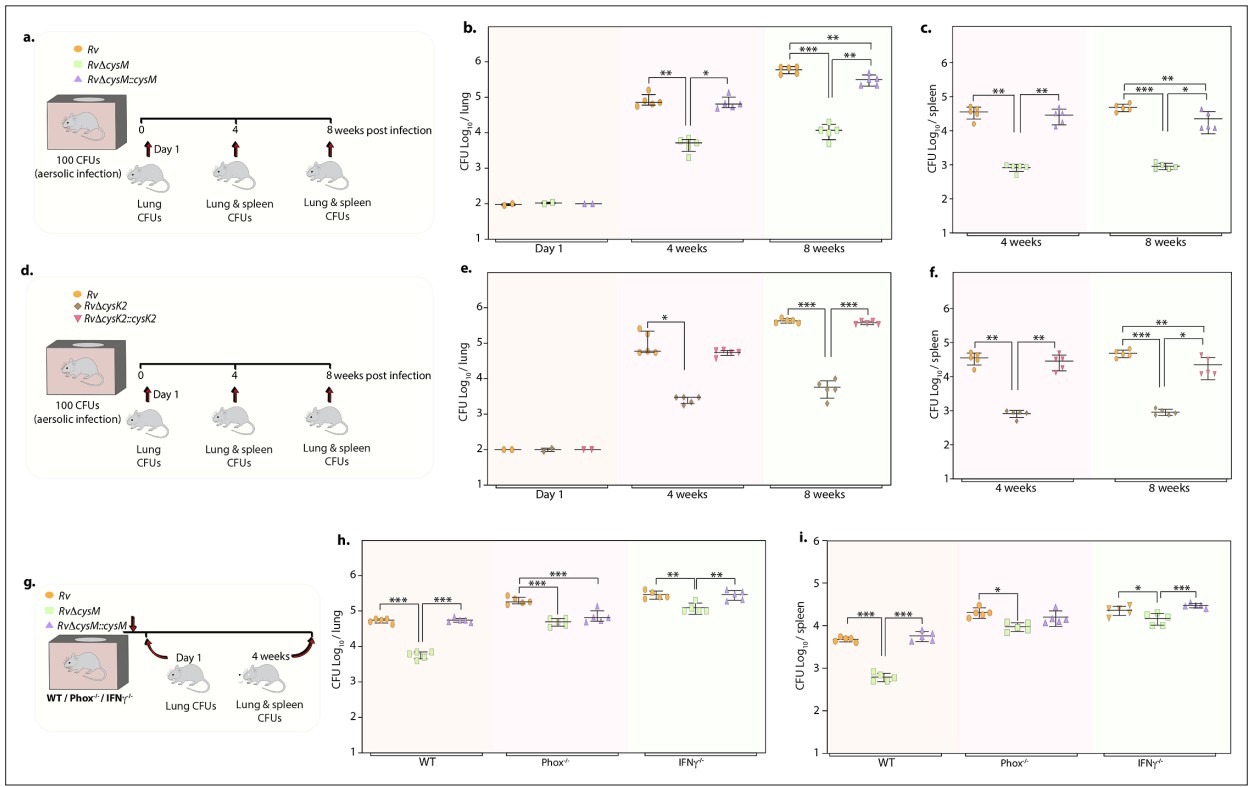

**Figure 5.** Deletion of non-canonical L-cysteine synthases attenuates mycobacterial survival in murine lungs and spleen. (**a**) Schematic outline of murine infection experiment. C57BL/6 ($n = 12$ per group) were infected with *Rv*, *RvΔcysM*, or *RvΔcysM::cysM* strains via an aerosol route. Colony-forming unit (CFU) was enumerated at day 1 ($n = 2$), week 4 ($n = 5$), and week 8 ($n = 5$). Each data point represents $\log_{10}$ CFU in lung (**b**) and spleen (**c**) of an infected animal, and the error bar indicates the median with interquartile range for each group. (**d**) Schematic outline of murine infection experiment. C57BL/6 ($n = 12$ per group) were infected with *Rv*, *RvΔcysK2*, or *RvΔcysK2::cysK2* strains via an aerosol route. CFU was enumerated at day 1 ($n = 2$), week 4 ($n = 5$), and week 8 ($n = 5$). Each data point represents $\log_{10}$ CFU in the lung (**e**) and spleen (**f**) of an infected animal, and the error bar indicates the median with interquartile range for each group. (**g**) Schematic outline of murine infection experiment. C57BL/6 ($n = 7$ per group), phox$^{-/-}$ ($n = 7$ per group), and IFNγ$^{-/-}$ ($n = 7$ per group) were infected with *Rv*, *RvΔcysM*, or *RvΔcysM::cysM* strains via an aerosol route. CFU was enumerated at day 1 ($n = 2$) and week 4 ($n = 5$). Each data point represents $\log_{10}$ CFU in the lung (**h**) and spleen (**i**) of an infected animal, and the error bar indicates the median with interquartile range for each group. (**a–i**) Statistical significance was drawn in comparison with *Rv* using one-way analysis of variance (ANOVA) followed by a post hoc test (Tukey test; GraphPad prism). *$p<0.05$, **$p<0.005$, ***$p < 0.0005$.

The online version of this article includes the following source data for figure 5:

**Source data 1.** CFU/ml values for *Figure 5*.

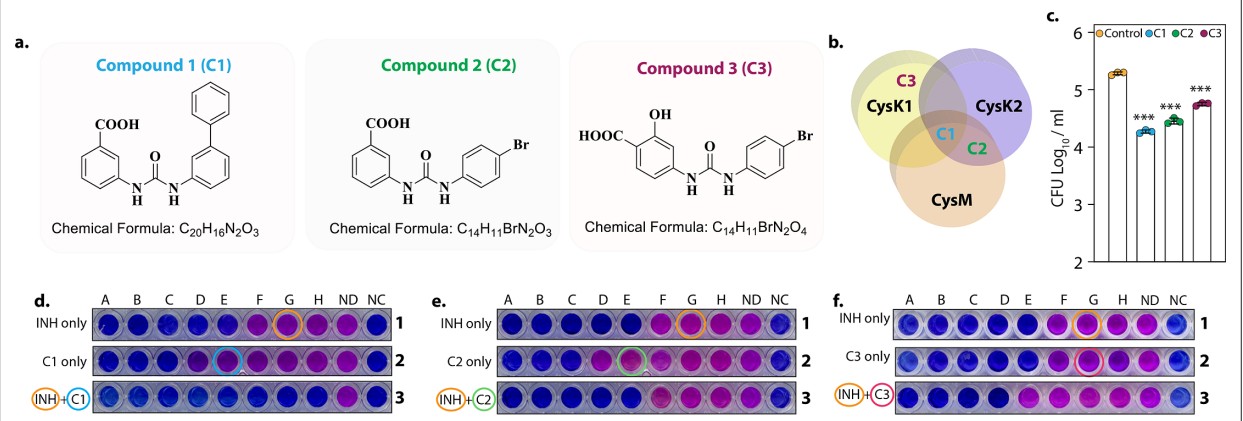

**Figure 6.** L-Cysteine synthases inhibitors can effectively kill Mtb within the host. Chemical structure and formula of lead compounds. (**b**). Venn diagram illustrating the specificity of compounds on mycobacterial L-cysteine synthases. (**c**). Peritoneal macrophages were infected with *Rv*. Four hours post-infection, cells were either left untreated (control) or 5 mg/ml C1 (**a**), C2 (**b**), or C3 (**c**) was added to the infected cells. Bar graphs represent mean $\log_{10}$ CFU/ml ± standard deviation (SD), and values from independent replicates are represented by individual data points (*n* = 3). Alamar blue assay is used to determine MIC values of C1 (**d**), C2 (**e**), and C3 (**f**) with and without Isoniazid in *Rv*. Starting concentration of INH in 1A is 0.96 µg/ml, 1B – 0.43 µg/ml and each subsequent column twofold dilution. Concentration of C1, C2, and C3 were 2.4 mg/ml in 2A and 1.2 mg/ml in 2B and in each subsequent column twofold dilution. The concentration of INH + C1 or C2 in 3 A was 0.015 µg/ml + 0.15 mg/ml, and 3B was 0.0075 µg/ml + 0.075 mg/ml. and each subsequent column twofold dilution. The concentration of INH + C3 in 3A was.015 µg/ml + 37 µg/ml, and 3B was 0.0075 µg/ml + 18.5 and each subsequent column twofold dilution. ND – no drug and NC – no culture. ***$p<0.0005$.

The online version of this article includes the following source data for figure 6:

**Source data 1.** CFU/ml values for *Figure 6c*.

**Source data 2.** Unedited and labelled image for *Figure 6d–f*.

**Source data 3.** Raw unedited for *Figure 6d–f*.

in *Figure 5h, i*, the survival of *RvΔcysM* is impaired in WT mice that produce both ROS and RNS, compared with phox$^{-/-}$ and IFNγ$^{-/-}$ mice. The survival defect observed due to the absence of CysM in the lungs was partially rescued in phox$^{-/-}$ mice. Due to the lack of ROS and RNS in IFNγ$^{-/-}$ mice, *RvΔcysM* showed higher bacillary load than in phox$^{-/-}$ mice (*Figure 5h*). *RvΔcysM* displayed better survival in phox$^{-/-}$ and IFNγ$^{-/-}$ mice in the spleen than lungs, which could either be because of relatively lesser ROS and RNS stress or imperiled response in clearing *Mtb* infection (*Figure 5i*). In contrast to the peritoneal macrophage infection experiment (*Figure 4*), attenuated survival of *RvΔcysM* was not completely salvaged in IFNγ$^{-}/^{-}$ mice, suggesting that CysM may also be involved in alleviating additional stresses such as nutrient deprivation and/or IFNγ independent ROS/RNS produced by the host. Data presented suggest that L-cysteine produced through non-canonical pathways are independently important for mycobacterial survival in vivo.

## L-Cysteine synthase inhibitors can effectively kill *Mtb* within the host

Data presented above demonstrated that CysM and CysK2 might serve as clinically important targets for adjunct TB therapy. Brunner et al. screened a compound library to identify inhibitors of mycobacterial L-cysteine synthases – CysK1, CysK2, and CysM. We selected three compounds – Compound 1 (C1) (named compound 2 in *Brunner et al., 2017*), which inhibits all three synthases; Compound 2 (C2) (compound 6 in *Brunner et al., 2017*), which inhibits both non-canonical L-cysteine synthases and Compound 3 (C3) (compound 31 in *Brunner et al., 2017*), which selectively inhibits CysK1 (*Figure 6a, b*). To examine the therapeutic potential of these inhibitors, we first tested their effect on the survival of *Rv* within host peritoneal macrophages. Treatment with C1 resulted in ~1 $\log_{10}$ attenuation compared with the untreated cells. Killing mediated by C2 or C3 was marginally lower compared with C1 (*Figure 6c*). Next, we examined the ability of these drugs to enhance the bactericidal activity of INH (isoniazid), a drug whose activity depends on redox state of the cell (*Vilchèze et al., 2005*), and therefore could be affected by disruption of L-cysteine downstream pathways. Towards this goal, we first measured the MICs (Minimal Inhibitory Concentration) of INH (isoniazid) (0.06 µg/ml) and C1 (0.6 mg/ml), C2 (0.6 mg/ml), and C3 (0.15 mg/ml) independently. As expected, all the compounds

were ineffective in killing *Mtb* because of redundant roles and relatively less requirement of L-cysteine synthases during regular growth conditions. To examine the combinatorial effect, we selected sub-MIC values of INH (0.03 µg/ml, encircled orange) and increasingly twofold diluted concentrations of C1, C2, or C3, all below MIC values (*Figure 6d–f*; blue encircled denotes the starting concentration) and assessed bacterial survival upon combinatorial treatment with INH and the three L-cysteine synthase inhibitors. The addition of either of the inhibitors below MIC rendered *Mtb* highly susceptible to INH, highlighting the potency of these compounds to serve as an adjunct therapy to combat mycobacterial infection.

## Discussion

Throughout its course of infection, *Mtb* must persist in a hostile, oxidizing, and nutrient-deprived environment of host macrophage. Understanding the dynamic metabolic interactions between the pathogen and its host is imperative to identify its weaknesses which can be exploited to design new-age chemotherapy. Our study provides convincing evidence that the genes involved in the L-cysteine biosynthetic pathway are attractive targets for the design of anti-mycobacterial drugs. Genes involved in sulfur assimilation and L-cysteine biosynthesis were found to be upregulated in the transcriptome profile of *Rv* subjected to various host-like stresses. The sulfur metabolism pathway was particularly enriched upon the addition of oxidizing agent, CHP (*Figure 1*, *Figure 1—figure supplements 2 and 3*). This observation has been consistently reported by multiple studies demonstrating the upregulation of sulfur assimilation and L-cysteine biosynthetic genes in response to oxidative stress, nutrient deprivation, and macrophage infection (*Hampshire et al., 2004*; *Manganelli et al., 2002*; *Pinto et al., 2004*; *Schnappinger et al., 2003*; *Khan et al., 2021*). Besides being involved in protein synthesis, L-cysteine is also important for the biosynthesis of L-methionine, *S*-adenosyl methionine, coenzyme A, and iron–sulfur clusters. As a thiol-containing molecule, L-cysteine contributes to the intracellular redox state directly and through the production of major antioxidants like mycothiol and ergothioneine. These numerous essential functions of L-cysteine prompted us to further examine its roles in the context of *Mtb* cellular metabolism and virulence.

Unlike *MtbΔcysH*, which was reported to be an L-cysteine auxotroph and thus required L-methionine or glutathione supplementation to grow in vitro (from which L-cysteine can be generated catabolically) (*Senaratne et al., 2006*), neither deletion of *cysK2* nor *cysM* impacted *Mtb* growth kinetics in vitro suggesting that these genes are functionally redundant during rich growth conditions (*Khan et al., 2021*; *Sao Emani et al., 2022*; *Figure 1—figure supplement 5*). Interestingly, the induction of various host-like stresses attenuated the growth of these mutants compared to the parental strain, pointing towards the possibility of an enhanced requirement of L-cysteine-derived antioxidants and other biomolecules to subvert these stresses (*Figure 1*). In agreement with this hypothesis, we previously showed that the cellular thiol levels are higher during oxidative stress in *Mtb* (*Khan et al., 2021*). Similarly, various MSH (reduced mycothiol) and ERG (ergothioneine) mutants display enhanced sensitivity to oxidative stress caused by treatment with $H_2O_2$, CHP, or $O_2^{\bullet-}$ (*Buchmeier et al., 2006*; *Buchmeier et al., 2003*; *Rawat et al., 2002*; *Rawat et al., 2007*; *Saini et al., 2016*).

While the transcriptomic profiles of mutants were highly similar to each other during regular growth conditions, the addition of CHP resulted in differential expression of >30% *Mtb* genes, indicating that cues and downstream effects of the two non-canonical L-cysteine synthases are partially non-overlapping, further underlying their importance for the bacillary growth inside the host (*Figure 2*). We also found that the steady-state levels (pool sizes) of key L-cysteine-derived antioxidants, ergothioneine, and mycothiol, are significantly different at 24 hr (*Figure 3*). The rate of synthesis of mycothiol, mycothione, and ergothioneine were observed to be lower in *RvΔcysM* compared to *Rv* and *RvΔcysK2*.

The addition of L-cysteine in oxidatively stressed cells nullified the compromised survival of the mutants indicating that *Mtb* cells are able to uptake L-cysteine from the extracellular medium, as shown previously (*Khan et al., 2021*), and reduced levels of L-cysteine in the mutants are chiefly responsible for their attenuated growth in the presence of stresses (*Figure 4*). Various *Mtb* mutants in mycobacterial sulfur metabolism genes are severely compromised to persist within the host and cause disease (*Senaratne et al., 2006*; *Sassetti et al., 2001*; *Curran and Hill, 1992*; *Sareen et al., 2003*; *Huet et al., 2005*; *Huet et al., 2006*; *Buchmeier and Fahey, 2006*). Inorganic sulfate is present at 300–500 µM in human plasma (*Buchmeier and Fahey, 2006*). However, the inability of host-derived sulfur/L-cysteine to compensate for attenuated survival of mutants is suggestive of either inaccessibility

of sufficient L-cysteine in *Mtb* niches or inefficient expression, function, or uptake by transporters in vivo. Importantly, compromised survival of *RvΔcysM* and *RvΔcysK2* in vivo was largely mitigated in IFNg–/– and Phox–/–, indicating that the non-canonical L-cysteine synthases, CysM and CysK2, independently facilitate mycobacterial survival during immune-mediated redox stress (*Figure 5*).

Using network analysis of the *Mtb* protein interactome, a flux balance analysis of the reactome and randomized transposon mutagenesis data, along with sequence analyses and a structural assessment of targetability, *Raman et al., 2008* reported CysK2 and CysM as high confidence drug targets. Similarly, inhibitors of CysM showed mycobactericidal activity in a nutrient-starvation model of dormancy (*Brunner et al., 2016*). Interestingly, humans do not reduce sulfur to produce L-cysteine; they rather synthesize L-cysteine through SAM-dependent transmethylation followed by transsulfuration of L-methionine. Owing to their complete absence in humans, mycobacterial L-cysteine biosynthetic genes and their regulators represent unique, attractive targets for therapeutic intervention (*Schnell et al., 2015*). We found that both *RvΔcysM* and *RvΔcysK2* displayed enhanced antibiotic sensitivity in vitro and within the host (data not shown). Importantly, a combination of L-cysteine synthase inhibitors with front-line TB drugs like INH, significantly reduced the bacterial survival in vitro (*Figure 6*). Altogether, this study demonstrates for the first time that the two non-canonical L-cysteine synthases have non-redundant biological functions. Deletion or biochemical inhibition of CysM or CysK2 perturbs redox homeostasis of *Mtb* and allow for the maximal effect of host macrophages antibacterial response, and thus increased elimination of virulent *Mtb*.

# Materials and methods

Key resources table

| Reagent type (species) or resource | Designation | Source or reference | Identifiers | Additional information |
|---|---|---|---|---|
| Gene (*Mycobacterium tuberculosis*) | CysM | Mycobrowser | CysM (Uniprot ID P9WP53) | |
| Gene (*Mycobacterium tuberculosis*) | CysK1 | Mycobrowser | CysK1 (Uniprot ID P9WP55) | |
| Gene (*Mycobacterium tuberculosis*) | CysK2 | Mycobrowser | CysK2(Uniprot ID Q79FV4) | |
| Strain, strain background (*Mycobacterium tuberculosis*) | H37Rv | ATCC | | |
| Strain, strain background (*Mycobacterium tuberculosis*) | RvDcysM | https://doi.org/10.15252/embj.2020106111 | | |
| Strain, strain background (*Mycobacterium tuberculosis*) | RvDcysK2 | This paper | | |
| Chemical compound, drug | C1 | *Brunner et al., 2017*. | | |
| Chemical compound, drug | C2 | *Brunner et al., 2017* | | |
| Chemical compound, drug | C3 | *Brunner et al., 2017* | | |
| Chemical compound, drug | Cumene hydroperoxide | Sigma Millipore | Cat no.: 247502 | |
| Chemical compound, drug | Isoniazid | Sigma Millipore | Cat no. I3377 | |

## Bacterial strains and culturing conditions

*Mtb* culturing conditions were performed as described previously (*Khan et al., 2021*; *Khan et al., 2017*).

## Generation of *RvΔcysK2* mutant and complementation strain

We generated gene replacement mutants through the recombineering method (*van Kessel and Hatfull, 2007*) as previously described (*Khan et al., 2021*; *Khan et al., 2017*). Briefly, 671 bp upstream of 176th nucleotide from 5′ end (5′ flank) and 643 bp downstream of 943rd nucleotide from 3′ end (3′ flank) of *cysK2* were PCR amplified from *Rv* genomic DNA using Phusion DNA polymerase (Thermo Scientific). The amplicons were digested with BstAPI and ligated with compatible hygromycin

resistance (*hyg^r*) cassette and *oriE*+cos $\lambda$ fragments (*van Kessel and Hatfull, 2007*), to generate the allelic exchange substrate (AES). AES was digested with SnaBI to release the LHS-*hyg^r*-RHS fragment, and the eluted fragment was electroporated into the recombineering proficient *Rv-ET* strain (*Khan et al., 2021*). Multiple *hyg^r* colonies were examined by PCRs to screen for legitimate recombination at *cysK2* locus. *RvΔcysK2*, thus generated, was cured of pNit-ET through negative selection on LB Agar plates containing 2% sucrose. To generate the complementation strain, full-length *cysK2* gene was PCR amplified from genomic DNA isolated from *Rv* as the template and Phusion DNA polymerase (Thermo Fischer Scientific). The amplicon was digested with NdeI-HindIII (NEB), cloned into the corresponding sites in pNit-3F and the resultant plasmid was electroporated into *RvΔcysK2* to generate the *RvΔcysK2::cysK2* strain. *RvΔcysM* and *RvΔcysM::cysM* strains were generated in the lab previously (*Khan et al., 2021*) using a similar method (*Manganelli et al., 2002*).

### Ex vivo infection experiments

Balb/c, C57BL/6 (B6), phox^−/− (B6.129S6-Cybbtm1Din/J; JAX# 002365) or IFN-γ^−/− (B6.129S7-Ifngtm1Ts/J; JAX#002287) mice were procured from The Jackson Laboratory. 4% thioglycolate (Hi-Media) was injected into the peritoneum cavity of 4- to 6-week-old mice and peritoneal macrophages were extracted 4 days post-injection and seeded and processed as described (*Khan et al., 2021*). In specific cases, the peritoneal macrophages were treated with 10 ng/ml IFNγ (BD Biosciences) overnight and 10 ng/ml lipopolysaccharide (LPS, Sigma) for 2 hr for activation. Where indicated, cells were further pretreated with 100 μM 1400 W (Sigma) overnight to inhibit iNOS or pretreated with 0.2 mM L-cysteine before infection. Single-cell *Mtb* suspensions were used for infection at 1:10 (host cells: bacteria) MOI. Four hours p.i, cells were washed thrice and replenished with complete RPMI (Roswell Park Memorial Institute 1640 medium) containing IFNγ + LPS, 1400 W, or L-cysteine, as required. The infected host cells were washed thrice with PBS, lysed using 0.05% SDS salt, and serial dilutions were plated on 7H11-OADC to enumerate bacillary survival.

### In vivo infection experiments

Mice (4–6 weeks) housed in ventilated cages at the Tuberculosis Aerosol Challenge Facility at the International Centre for Genetic Engineering and Biotechnology (New Delhi, India) were infected via aerosol route with ~100 bacilli using the Madison Aerosol Chamber (University of Wisconsin, Madison, WI). At 24 hr p.i, mice (*n* = 2; per group) were euthanized to determine the bacterial deposition. At 4/8 weeks p.i, lungs and spleen were homogenized and plated on 7H11+OADC containing PANTA.

### RNA isolation and qRT-PCRs (quantitative reverse transcriptase-polymerase chain reactions)

*Mtb* strains were cultured in triplicates in 7H9-ADS till OD (Optical Density) reached 0.3–0.4. One set was left untreated (control), the other was treated with 50 μM of CHP for 6 hr. A culture equivalent to 10 $OD_{600}$ was resuspended in TRIzol (Invitrogen). Zirconium beads were added to the cell-Trizol mix to facilitate lysing *Mtb* cells with the help of a bead beater (MP FastPrep system, MP Biomedicals) and RNA was extracted and analyzed as described (*Khan et al., 2021*). Data were plotted as $2^{(-\Delta\Delta Ct)}$ wherein the gene expression was normalized with respect to 16s rRNA (*rrs* gene), followed by normalization with control strain/condition/group.

### RNA-seq and analysis

Total RNA was isolated from two to three biological replicates of indicated *Mtb* strains, and their concentrations were checked by Qubit (Thermo Fischer Scientific) followed by quality assessment through Agilent 2100 BioAnalyzer (Agilent RNA 6000 Nano Kit). Both sets of RNA seq represented in *Figures 1 and 5* were processed and run at the same time, the same set of control (*Rv*) triplicates were used for the analysis of both figures. Samples with RIN values >7 were processed for RNA-seq using the Illumina NovaSeq 6000 Platform (CSIR-CCMB central facility; read length of 100 bp, 20 million paired-end reads/sample).

Illumina adapters and low-quality reads were discarded and those with quality scores <20 and smaller than 36 bp were eliminated from raw sequencing reads using cutadapt (*Martin, 2011*). Processed reads were mapped to the *Mtb* H37Rv (see here), using hisat2 with default parameters (*Kim et al., 2019*). Uniquely aligned reads were counted with the help of feature Counts of Subread

package (*Liao et al., 2014*) and those with total read count <10 across all the samples were removed, and the rest were used for further downstream analysis. DEGs were identified using DESeq2 (*Love et al., 2014*) and those with adjusted p-value <0.05 and absolute $\log_2$ Fold change >1 or 0.5 were considered. The raw read counts were rlog normalized the raw read counts for PCA plot and heat map with the DESeq2 package.

### Functional enrichment analysis

Functional enrichment analysis was performed with DAVID web services (*Sherman et al., 2022*). We specifically used GO terms and KEGG (Kyoto Encyclopedia of Genes and Genomes) pathways for this analysis. Only top 10 enriched hits based on gene counts were plotted.

### Preparation of samples for metabolomics

*Mtb* strains were grown in 7H9 media until an OD 1 and then inoculated onto 0.22 µm nitrocellulose filters and grown on 7H10 plates containing ADS (0.5 g/l Bovine Serum Albumin Fraction V, 0.2% dextrose, and 0.085% NaCl) for 5 days. The filters were then transferred to 7H10 plus ADS plates containing sodium sulfate-$^{34}$S (Merck 718882) containing either 50 µM CHP or no CHP for 24 and 48 hr. The metabolites were extracted by mechanical lysis in cold acetonitrile/methanol/water (2:2:1) containing 0.1 mm acid washed Zirconia beads. The lysates were clarified by centrifugation and filtered through a 0.22 µm Spin-X column (Costar). The lysates were mixed 1:1 with acidified acetonitrile (0.1% formic acid). Due to the tendency of *M. tuberculosis* to form clamps, which significantly skew any cell number estimations we normalized samples to protein/peptide concentration using the BCA assay kit (Thermo). Therefore, our liquid chromatography–mass spectrometry (LC–MS) data are express as ion counts/mg protein or ratios of that for the same metabolite. This is a standard way to express ion abundance data (*Mougous et al., 2002*; *Pinto et al., 2007*).

### LC–MS metabolomics

LC–MS analysis was done in an Agilent 1290 Infinity II HPLC connected to a 6230B time-of-flight (ToF) mass spectrometer using a Dual AJS ESI ionization source. Compounds were separated in a Cogent Diamond Hydride Type C silica column (2.1 × 150 mm). Solvent A was LC–MS grade $H_2O$ + 0.1% (vol/vol) formic acid and solvent B was acetonitrile +0.1% (vol/vol) formic acid. The gradient was from 85% B to 5% B over 14 min. Flow rate was 0.4 ml/min. The ion source parameters were as follows: gas temperature 250°C, drying gas flow rate 13 l/min, nebulizer pressure 35 psig, sheath gas temperature 350°C, sheath gas flow 12 l/min, capillary voltage 3500 V, and nozzle voltage 2000 V. The ion optic voltages were 110 V for the fragmentor, 65 V for the skimmer, and 750 for the octopole radio frequency voltage. MS data were analyzed with the MassHunter suite version B0.7.0.00.

### MIC analysis

MIC values were assessed using Alamar Blue assay, as described previously (*Khan and Nandicoori, 2021*). Briefly, 100 µl 7H9-ADS medium without Tween 80 was added to each well of a 96-well plate. First well of each column were filled with 100 µl of the test drug/antibiotic, which was then serially diluted across the column. *Mtb* cells corresponding to 0.01 $A_{600}$ were diluted in 100 µl 7H9-ADS medium and were added to each well. Two rows one in which the drug was not added (no drug), and the other wherein *Mtb* cells were replaced with 7H9-ADS (no cells) acted as controls. The 96-well plate was sealed with parafilm and kept at 37°C. After 5 days, 20 µl of 0.25% filter-sterilized resazurin was added to each well, and colour development was captured after 24 hr.

### Statistical analysis

Unless otherwise specified, experiments were performed in triplicates and repeated independently at least twice. CFU results were plotted, and significance of the datasets was calculated using one-way analysis of variance followed by a post hoc test (Tukey test) on GraphPad Prism 5. Figures were customized using Adobe Illustrator version 26.3.1. Statistical significance was set at p-values <0.05 significant (*p <0.05; **p <0.005; ***p <0.0005). Source datasets can be requested from the corresponding author.

## Acknowledgements

We thank Dr. Apruva Sarin for phox$^{-/-}$ mice and thoughtful discussion; ICGEB for access to their TACF; CCMB for access to their sequencing facility. MZK was supported by Research Associateship from SERB (CRG/2018/001294). Research reported in this publication was supported by the core grant of the National Institute of Immunology; DBT grant BT/PR13522/COE/34/27/2015; SERB grant CRG/2018/001294, and JC Bose fellowship JCB/2019/000015 to VKN. Work in LPSC's laboratory was supported by the Francis Crick Institute, which receives its core funding from Cancer Research UK Grant FC001060, UK MRC Grant FC001060, and Wellcome Trust Grant FC001060.

## Additional information

### Funding

| Funder | Grant reference number | Author |
|---|---|---|
| Science and Engineering Research Board | CRG/2018/001294 | Mehak Zahoor Khan Vinay Kumar Nandicoori |
| Science and Engineering Research Board | JCB/2019/000015 | Vinay Kumar Nandicoori |
| Department of Biotechnology, Ministry of Science and Technology, India | BT/PR13522/COE/34/27/2015 | Vinay Kumar Nandicoori |
| Cancer Research UK | FC001060 | Luiz Pedro S de Carvalho |
| Medical Research Council | FC001060 | Luiz Pedro S de Carvalho |
| Wellcome Trust | FC001060 | Luiz Pedro S de Carvalho |

The funders had no role in study design, data collection, and interpretation, or the decision to submit the work for publication. For the purpose of Open Access, the authors have applied a CC BY public copyright license to any Author Accepted Manuscript version arising from this submission.

### Author contributions

Mehak Zahoor Khan, Conceptualization, Formal analysis, Investigation, Methodology, Writing – original draft, Writing – review and editing; Debbie M Hunt, Formal analysis, Investigation; Biplab Singha, Investigation, Methodology; Yogita Kapoor, Investigation, Methodology, Writing – review and editing; Nitesh Kumar Singh, Formal analysis; D V Sai Prasad, Sriram Dharmarajan, Resources; Divya Tej Sowpati, Formal analysis, Writing – review and editing; Luiz Pedro S de Carvalho, Conceptualization, Formal analysis, Supervision, Methodology, Writing – review and editing; Vinay Kumar Nandicoori, Conceptualization, Supervision, Funding acquisition, Investigation, Writing – original draft, Project administration, Writing – review and editing

### Author ORCIDs

Yogita Kapoor ⓘ https://orcid.org/0000-0001-8543-1755
Luiz Pedro S de Carvalho ⓘ https://orcid.org/0000-0003-2875-4552
Vinay Kumar Nandicoori ⓘ https://orcid.org/0000-0002-5682-4178

### Ethics

Protocols for animal experiments were preapproved by the Animal Ethics Committee of the National Institute of Immunology, New Delhi, India (IAEC numbers 409/16 and 462/18) as per standard institutional guidelines.

Reviewer #1 (Public Review): https://doi.org/10.7554/eLife.91970.3.sa1
Author response https://doi.org/10.7554/eLife.91970.3.sa2

# Additional files

## Supplementary files
• Supplementary file 1. Differentially expressed genes (DEGs) analysis *Rv* under stress vs. *Rv*.

• Supplementary file 2. We compiled a list of differentially expressed genes in at least one of the five comparisons against control.

• Supplementary file 3. Pathway enrichment analysis of differentially expressed genes (DEGs) from *Supplementary file 1*.

• Supplementary file 4. Differentially expressed gene (DEG) analysis of RvΔcysM or RvΔcysK2 vs. Rv with or without cumene hydroperoxide (CHP) stress and DEG analysis RvΔcysM + CHP vs. RvΔcysK2 + CHP.

• Supplementary file 5. Pathway enrichment analysis of differentially expressed genes (DEGs) from *Supplementary file 3*.

• MDAR checklist

## Data availability
RNA-seq data are available at the NCBI Gene Expression Omnibus Database, accession no. GEO GSE225792, the link to the database is https://www.ncbi.nlm.nih.gov/geo/query/acc.cgi?acc=GSE225792.

The following dataset was generated:

| Author(s) | Year | Dataset title | Dataset URL | Database and Identifier |
|---|---|---|---|---|
| Vinay N, Mehak Zahoor K, Biplab S, Nitesh Kumar S, Tej Divya S | 2023 | Divergent downstream biosynthetic pathways are supported by L-cysteine synthases of Mycobacterium tuberculosis | https://www.ncbi.nlm.nih.gov/geo/query/acc.cgi?acc=GSE225792 | NCBI Gene Expression Omnibus, GSE225792 |

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
