## [Editor Report · eLife assessment]

Sulphur atoms derived from cysteine are thought to play significant roles in maintaining redox homeostasis in Mycobacterium tuberculosis, which encounters stresses associated with immune cell interactions. In this **valuable** manuscript, the authors provide **solid** evidence that the genes encoding cysteine biosynthetic enzymes (cysM and cysK2) are required to maintain full viability of M. tuberculosis under in vitro stress conditions, macrophage infections, and within the lung tissues of mice. The manuscript presents transcriptomic and metabolomic evidence to support the hypothesis that CysM and CysK2 play distinct roles in maintaining cysteine-derived metabolite pools under stress conditions. The work will be of interest to microbiologists in general.

---

## [Referee Report · Reviewer #1 (Public Review)]

Summary:

Khan et. al., investigated the functional redundancy of the non-canonical L-cysteine synthases of M. tuberculosis, CysM and CysK2, focussing on their role in mitigating the effects of host-derived stress. They found that while deletion mutants of the two synthases (Rv∆cysM, Rv∆cysK2) have similar transcriptomes under standard conditions, their transcriptional response to oxidative stress is distinct. The impact of deleting the synthases also differentially affected the pools of L-cysteine-derived metabolites. They show that the mutants (Rv∆cysM, Rv∆cysK2) have impaired survival in peritoneal macrophages and in a mouse model of infection. Importantly, they show that the survival of the mutants increases when the host are defective in producing reactive oxygen and nitrogen species, linking the phenotype to a defect in combating host-derived stress. Finally, they show that compounds inhibiting L-cysteine synthases reduces intracellular survival of M. tuberculosis.

Strengths:

(1) The distinct transcriptome of the Rv∆cysM and Rv∆cysK2 mutants in the presence of oxidative stress provides solid evidence that these mutants are distinct in their response to oxidative stress, and suggests that they are not functionally redundant.

(2) The use of macrophages from phox-/- and INF-/- mice and an iNOS inhibitor for the intracellular survival assays provides solid evidence that the survival defect seen for the Rv∆cysM and Rv∆cysK2 mutants is related to their reduced ability to combat host-derive oxidative and nitrosative stress. This is further supported by the infection studies in phox-/- and INF-/- mice.

Weaknesses:

Inclusion of the complemented strains in the metabolite study would strengthen the data. Furthermore, using an alternate method to quantify the MSH:MSSM ratio would provide insight into the redox homoeostasis in mutants in the presence and absence of CHP to support the statement that "deletion or inhibition of CysM or CysK2 perturbs redox homeostasis of Mtb".

The authors sought to investigate the functional redundancy of the non-canonical L-cysteine synthases CysM and CysK2. While their distinct transcriptional response to oxidative stress suggests distinct physiological roles, the study did not explore these differences, and therefore provides only preliminary insight into the underlying reasons for this observation. In the context of drug development, this work suggests that while L-cysteine synthases inhibitors do not have high potency for killing intracellular M. tuberculosis, they have potential for decreasing the pathogen's survival in the presence of host-derive stress.

---

## [Author Response]

The following is the authors’ response to the original reviews.

**Public Reviews:**

**Reviewer #1 (Public Review):**
Summary:Khan et. al., investigated the functional redundancy of the non-canonical L-cysteine synthases of M. tuberculosis, CysM and CysK2, focussing on their role in mitigating the effects of host-derived stress. They found that while deletion mutants of the two synthases (Rv∆cysM, Rv∆cysK2) have similar transcriptomes under standard conditions, their transcriptional response to oxidative stress is distinct. The impact of deleting the synthases also differentially affected the pools of L-cysteinederived metabolites. They show that the mutants (Rv∆cysM, Rv∆cysK2) have impaired survival in peritoneal macrophages and in a mouse model of infection. Importantly, they show that the survival of the mutants increases when the host is defective in producing reactive oxygen and nitrogen species, linking the phenotype to a defect in combating host-derived stress. Finally, they show that compounds inhibiting L-cysteine synthases reduce the intracellular survival of M.tuberculosis.Strengths:(1) The distinct transcriptome of the Rv∆cysM and Rv∆cysK2 mutants in the presence of oxidative stress provides solid evidence that these mutants are distinct in their response to oxidative stress, and suggests that they are not functionally redundant.(2) The use of macrophages from phox-/- and INF-/- mice and an iNOS inhibitor for the intracellular survival assays provides solid evidence that the survival defect seen for the Rv∆cysM and Rv∆cysK2 mutants is related to their reduced ability to combat host-derive oxidative and nitrosative stress. This is further supported by the infection studies in phox-/- and INF-/- mice.Weaknesses:(1) There are several previous studies looking at the transcriptional response of M. tuberculosis to host-derived stress, however, the authors do not discuss initial RNA-seq data in the context of these studies. Furthermore, while several of the genes in sulfur assimilation and L-cysteine biosynthetic pathway genes are upregulated by more than one stress condition, the data does not support the statement that it is the "most commonly upregulated pathway in Mtb exposed to multiple host-like stresses".

We have made changes in the manuscript in line with reviewer’s suggestion.

“Thus RNA-Seq data suggest that genes involved in sulfur assimilation and L-cysteine biosynthetic pathway are upregulated during various host-like stresses in Mtb (Figure S2). Given the importance of sulphur metabolism genes in in vivo survival of Mtb [1, 2], it is not surprising that these genes are dynamically regulated by diverse environment cues. Microarray studies have shown upregulation of genes encoding sulphate transporter upon exposure to hydrogen peroxide and nutrient starvation [3-7] Similarly, ATP sulfurlyase and APS kinase is induced during macrophage infection and by nutrient depletion. Induction of these genes that coordinate first few steps of sulphur assimilation pathway indicate that probable increase in biosynthesis of sulphate containing metabolites that may be crucial against host inflicted stresses. Furthermore, genes involved in synthesis of reduced sulphur moieties (cysH, sirA and cysM) are also induced by hydrogen peroxide and nutrient starvation. Sulfur metabolism has been postulated to be important in transition to latency. This hypothesis is based on transcriptional upregulation of cysD, cysNC, cysK2, and cysM upon exposure to hypoxia. Multiple transcriptional profiling studies have reported upregulation of moeZ, mec, cysO and cysM genes when cells were subjected to oxidative and hypoxic stress [1, 6-11] further suggesting an increase in the biosynthesis of reduced metabolites such as cysteine and methionine and sulfur containing cell wall glycolipids upon exposure to oxidative stress [12]. We have modified the sentence to “significantly upregulated pathway in Mtb exposed to multiple host-like stresses”

(2) For the quantification of the metabolites, it isn't clear how the abundance was calculated (e.g., were standards for each metabolite used? How was abundance normalised between samples?), and this information should be included to strengthen the data.

Thanks for picking up this. We have extended our description of metabolomics methods. It now reads: “Due to the tendency of M. tuberculosis to form clamps, which significantly skews any cell number estimation we normalized samples to protein/peptide concentration using the BCA assay kit (Thermo). Therefore, our LC-MS data is expressed as ion counts/mg protein or ratios of that for the same metabolite. This is a standard way to express ion abundance data as it was done previously [13, 14].

Furthermore, labelling with L-methionine was performed to determine the rate of synthesis of the L-cysteine-derived metabolites. L-cysteine is produced from L-methionine via the transsulfuration pathway, which is independent of CysM and CysK2. It is therefore difficult to interpret this experiment, as the impact of deleting CysM and CysK2 on the transsulfuration pathway is likely indirect.

The reviewer may have misunderstood the experiment and the results presented. Labelling was not performed with L-methionine. We use 34S derived from SO42-, to monitor reductive assimilation of sulfur and its transit from S2- until L-methionine, passing through cysteine. We specified in material and methods that we have used sodium sulfate-34S (Merck 718882), as our label source of sulfur. This method was first employed in M. tuberculosis by the Bertozzi group to identify sulfolipids in mycobacteria. Therefore, we are not measuring transsulfuration, but instead direct synthesis of L-methionine via cysteine, and consequently we are indeed assessing the importance of cysK2 and cysM in this process. We have now added to the results section (page 9) that we employed (Na34SO4) for labeling, to make sure other readers will not think we are measuring transulfuration.

(3) The ability of L-cysteine to rescue the survival defect of the Rv∆cysM and Rv∆cysK2 mutants in macrophages is interpreted as exogenous L-cysteine being able to compensate for reduced intracellular levels. However, there is no evidence that L-cysteine is being taken up by the mutants and an alternate explanation is that L-cysteine functions as an antioxidant within cells i.e., it reduces intracellular ROS.

The concentration of L-cysteine used for peritoneal macrophage survival rescue experiments was titrated to have no minimum survival advantage in case of wild-type Rv. Thus, at the given concentration, we believe that the contribution of cysteine in reducing intracellular ROS within cells does not have a major role since there is no significant difference in the survival of wild-type Rv strain. Had cysteine reduced intracellular ROS, we would expect increased bacterial survival of Rv due to diminished oxidative stress.

Furthermore, L-cysteine addition also mitigates CHP induced survival defect in vitro [15] and nullifies observed effect of Cysteine inhibitors in vitro [16] suggesting that cysteine or cystine can be transported into Mtb. This has also been previously shown in case of AosR mutant strain [15], CysH [2] and over 70% uptake of exogenously added [35S] cysteine to a growing culture of Mtb [17].

The authors sought to investigate the functional redundancy of the non-canonical L-cysteine synthases CysM and CysK2. While their distinct transcriptional response to oxidative stress suggests distinct physiological roles, the study did not explore these differences and therefore provides only preliminary insight into the underlying reasons for this observation. In the context of drug development, this work suggests that while L-cysteine synthase inhibitors do not have high potency for killing intracellular M. tuberculosis, they have the potential to decrease the pathogen's survival in the presence of host-derive stress.
**Reviewer #2 (Public Review):**
Summary:The paper examines the role L-cysteine metabolism plays in the biology of Mycobacterium tuberculosis. The authors have preliminary data showing that Mycobacterium tuberculosis has two unique pathways to synthesize cysteine. The data showing new compounds that act synergistically with INH is very interesting.Strengths:RNAseq data is interesting and important.Weaknesses:The paper would be strengthened if the authors were to add further detail to their genetic manipulations.The authors provide evidence that they have successfully made a cysK2 mutant by recombineering. This data looks promising, but I do not see evidence for the cysM deletion. It is also important to state what sort of complementation was done (multicopy plasmid, integration proficient vector, or repair of the deletion). Since these mutants are the basis for most of the additional studies, these details are essential. It is important to include complementation in mouse studies as unexpected loss of PDIM could have occurred.

The details of CysM knockout generation have been previously published ([15]; Appendix Figure S4), and complementation strain details are provided in the methods section.

**Reviewer #3 (Public Review):**
In this work, the authors conduct transcriptional profiling experiments with Mtb under various different stress conditions (oxidative, nitrosative, low pH, starvation, and SDS). The Mtb transcriptional responses to these stress conditions are not particularly new, having been reported extensively in the literature over the past ~20 years in various forms. A common theme from the current work is that L-cysteine synthesis genes are seemingly up-regulated by many stresses. Thus, the authors focused on deleting two of the three L-cysteine synthesis genes (cysM and cysK2) in Mtb to better understand the roles of these genes in Mtb physiology.The cysM and cysK2 mutants display fitness defects in various media (Sautons media, starvation, oxidative and nitrosative stress) noted by CFU reductions. Transcriptional profiling studies with the cysM and cysK2 mutants revealed that divergent gene signatures are generated in each of these strains under oxidative stress, suggesting that cysM and cysK2 have non-redundant roles in Mtb's oxidative stress response which likely reflects the different substrates used by these enzymes, CysO-L-cysteine and O-phospho-L-serine, respectively. Note that these studies lack genetic complementation and are thus not rigorously controlled for the engineered deletion mutations.The authors quantify the levels of sulfur-containing metabolites (methionine, ergothioneine, mycothiol, mycothionine) produced by the mutants following exposure to oxidative stress. Both the cysM or cysK2 mutants produce more methionine, ergothioneine, and mycothionine relative to WT under oxidative stress. Both mutants produce less mycothiol relative to WT under the same condition. These studies lack genetic complementation and thus, do not rigorously control for the engineered mutations.Next, the mutants were evaluated in infection models to reveal fitness defects associated with oxidative and nitrosative stress in the cysM or cysK2 mutants. In LPS/IFNg activated peritoneal macrophages, the cysM or cysK2 mutants display marked fitness defects which can be rescued with exogenous cysteine added to the cell culture media. Peritoneal macrophages lacking the NADPH oxidase (Phox) or IFNg fail to produce fitness phenotypes in the cysM or cysK2 mutants suggesting that oxidative stress is responsible for the phenotypes. Similarly, chemical inhibition of iNOS partly abrogated the fitness defect of the cysM or cysK2 mutants. Similar studies were conducted in mice lacking IFNg and Phox establishing that cysM or cysK2 mutants have fitness defects in vivo that are dependent on oxidative and nitrosative stress.Lastly, the authors use small molecule compounds to inhibit cysteine synthases. It is demonstrated that the compounds display inhibition of Mtb growth in 7H9 ADC media. No evidence is provided to demonstrate that these compounds are specifically inhibiting the cysteine synthases via "ontarget inhibition" in the whole Mtb cells. Additionally, it is wrongly stated in the discussion that "combinations of L-cys synthase inhibitors with front-line TB drugs like INH, significantly reduced the bacterial load inside the host". This statement suggests that the INH + cysteine synthase inhibitor combinations reduce Mtb loads within a host in an infection assay. No data is presented to support this statement.

We agree with the reviewer that the experiments do not conclusively prove that these compounds specifically inhibit the cysteine synthases via "on-target inhibition" in the whole Mtb cells. However, the inhibitors used in this study have been previously profiled in vitro (https://www.sciencedirect.com/science/article/abs/pii/S0960894X17308405?via%3Dihub). We have modified the sentence to “a combination of L-cysteine synthase inhibitors with front-line TB drugs like INH, significantly reduced the bacterial survival in vitro”

References

(1) Hatzios, S.K. and C.R. Bertozzi, The regulation of sulfur metabolism in Mycobacterium tuberculosis. PLoS Pathog, 2011. 7(7): p. e1002036.

(2) Senaratne, R.H., et al., 5'-Adenosinephosphosulphate reductase (CysH) protects Mycobacterium tuberculosis against free radicals during chronic infection phase in mice. Mol Microbiol, 2006. 59(6): p. 1744-53.

(3) Betts, J.C., et al., Evaluation of a nutrient starvation model of Mycobacterium tuberculosis persistence by gene and protein expression profiling. Mol Microbiol, 2002. 43(3): p. 717-31.

(4) Hampshire, T., et al., Stationary phase gene expression of Mycobacterium tuberculosis following a progressive nutrient depletion: a model for persistent organisms? Tuberculosis (Edinb), 2004. 84(3-4): p. 228-38.

(5) Schnappinger, D., et al., Transcriptional Adaptation of Mycobacterium tuberculosis within Macrophages: Insights into the Phagosomal Environment. J Exp Med, 2003. 198(5): p. 693-704.

(6) Voskuil, M.I., et al., The response of mycobacterium tuberculosis to reactive oxygen and nitrogen species. Front Microbiol, 2011. 2: p. 105.

(7) Voskuil, M.I., K.C. Visconti, and G.K. Schoolnik, Mycobacterium tuberculosis gene expression during adaptation to stationary phase and low-oxygen dormancy. Tuberculosis (Edinb), 2004. 84(3-4): p. 218-27.

(8) Brunner, K., et al., Profiling of in vitro activities of urea-based inhibitors against cysteine synthases from Mycobacterium tuberculosis. Bioorg Med Chem Lett, 2017. 27(19): p. 4582-4587.

(9) Manganelli, R., et al., Role of the extracytoplasmic-function sigma factor sigma(H) in Mycobacterium tuberculosis global gene expression. Mol Microbiol, 2002. 45(2): p. 365-74.

(10) Burns, K.E., et al., Reconstitution of a new cysteine biosynthetic pathway in Mycobacterium tuberculosis. J Am Chem Soc, 2005. 127(33): p. 11602-3.

(11) Manganelli, R., et al., The Mycobacterium tuberculosis ECF sigma factor sigmaE: role in global gene expression and survival in macrophages. Mol Microbiol, 2001. 41(2): p. 423-37.

(12) Tyagi, P., et al., Mycobacterium tuberculosis has diminished capacity to counteract redox stress induced by elevated levels of endogenous superoxide. Free Radic Biol Med, 2015. 84: p. 344-354.

(13) de Carvalho, L.P., et al., Metabolomics of Mycobacterium tuberculosis reveals compartmentalized co-catabolism of carbon substrates. Chem Biol, 2010. 17(10): p. 1122-31.

(14) Agapova, A., et al., Flexible nitrogen utilisation by the metabolic generalist pathogen Mycobacterium tuberculosis. Elife, 2019. 8.

(15) Khan, M.Z., et al., Redox homeostasis in Mycobacterium tuberculosis is modulated by a novel actinomycete-specific transcription factor. EMBO J, 2021. 40(14): p. e106111.

(16) Brunner, K., et al., Inhibitors of the Cysteine Synthase CysM with Antibacterial Potency against Dormant Mycobacterium tuberculosis. J Med Chem, 2016. 59(14): p. 6848-59.

(17) Wheeler, P.R., et al., Functional demonstration of reverse transsulfuration in the Mycobacterium tuberculosis complex reveals that methionine is the preferred sulfur source for pathogenic Mycobacteria. J Biol Chem, 2005. 280(9): p. 8069-78.

**Recommendations For The Authors:**

**Reviewer #1 (Recommendations For The Authors):**
(1) In Figure S1 it would be useful to include the reverse transsulfuration pathway given that it contributes to the L-cysteine pool, and that L-methionine was used for metabolite labelling experiments.

We are in agreement with the reviewer’s suggestion, and we have included reverse transsulfuration in Fig S1. Please note that Labelling was not performed with L-methionine. We used 34S derived from SO42-to monitor the reductive assimilation of sulfur and its transit from S2- until Lmethionine, passing through cysteine. We specified in material and methods that we have used sodium sulfate-34S (Merck 718882), as our label source of sulfur. This method was first employed in M. tuberculosis by the Bertozzi group to identify sulfolipids in mycobacteria. Therefore, we are not measuring transsulfuration but instead a direct synthesis of Lmethionine via cysteine, and consequently, we are indeed assessing the importance of cysK2 and cysM in this process. We have now added to the results section (page 9) that we employed (Na34SO4) for labeling to make sure other readers will not think we are measuring transulfuration.

**Author response image 1. sa2fig1:** 

(2) In Figure S2 it is unclear why the control is included in this figure given that the stress conditions were compared to the control. What is the control being compared to here?

The heat maps of controls have been included to demonstrate relative gene expression in independent/each of the replicates. The normalized count for the differentially expressed genes are plotted. To better understand the RNA-seq results, we plotted the fold change of differentially expressed genes due to different stress conditions (New figure & table- Figure S3 & Table S2). This allowed us to understand the expression profile of genes in all the stress conditions simultaneously, regardless of whether they were identified as differentially expressed. The data revealed that specific clusters of genes are up- and downregulated in oxidative, SDS, and starvation conditions. In comparison, the differences observed in the pH 5.5 and nitrosative conditions were limited (Figure S3 & Table S2).

(3) In Figure S3 it would be more informative to show fold-enrichment than gene counts in (b) to (f).

In our opinion, gene counts are more informative when plotting GO enrichments, as the number of genes in each GO category can vary drastically. The significance values are already calculated based on the fold enrichment of a category compared to the background, and hence, p-adj values plotted on the x-axis can be sort of a proxy for fold enrichment. Hence, instead of plotting two related variables, plotting the total gene counts that belonged to a category is usually helpful for the reader in understanding the “scale” in which a category is affected.

(4) Figure 1c standard Sautons is a defined media, and is not nutrient-limiting - the authors should clarify the composition of the media that they used here.

The composition of Sautons media used in the study is 0.5g/L MgSO4.7H20, 2 g/L citric acid, 1g/L L-asparagine, 0.3 g/L KCl.H20, 0.2% glycerol, 0.64 g/L FeCl3, 100 μM NH4Cl and 0.7 g/L K2HPO4.3H20. We have modified the sentence in line with reviewer’s suggestion.

(5) The authors claim that the distinct transcriptomes for the two mutants indicate that "CysM and CysK2 distinctly modulate 324 and 1104 genes". The effect is likely due to distinct downstream consequences of the deletions, rather than direct regulation by the synthases. This section should be reworded for clarity.

We have modified the sentence in line with reviewer’s suggestion.

(6) In Figure 3 it would be useful to express mycothione levels as a percentage of the total mycothiol pool to give an indication of the extent to which the thiol is being oxidised.

While we appreciate reviewer’s suggestion, we cannot make ratios of IC for two different compounds, as they ionize different. 100 ion counts of one does NOT equal to 100 ion counts of the other.

(7) Figure 6 is difficult to interpret as the concentrations used in the INH + inhibitor wells are not clear. It would be useful to indicate the concentrations of each compound added next to the wells in the figure.

We have modified the figure and legends in line with reviewer’s suggestion

**Reviewer #2 (Recommendations For The Authors):**
(1) Document the cysM deletion.

The details of CysM knockout generation have been previously published ([15]; Appendix Figure S4), and complementation strain details are provided in the methods section.

(2) The oxidative stress CHP is not defined in the figure legend.

We have modified the legend in line with the reviewer’s suggestion.

(3) Can we see the structures of the compounds?

Kindly refer to Fig 6a for the structures of compounds

(4) Fix the genetics and the paper is very interesting.I might be missing something. The authors do provide promising complementation data for several of the stresses. Provide evidence for the cysM deletion and complementation and the data will be very compelling. The focus of the paper is important for our understanding of the biology of Mycobacterium tuberculosis.

Thank you for appreciating our study. The details of CysM knockout and complementation strain generation have been previously published ([15]; Appendix Figure S4 & Methods). CysK2 mutant and complementation strain details are included in the present manuscript (Figure 1b & Methods).

**Reviewer #3 (Recommendations For The Authors):**
The transcriptional profiling studies do not rigorously control for the engineered mutations using genetic complementation.

The complementation strains used in all in vitro, ex vivo and in vivo experiments showcase that the phenotypes associated with knockouts are gene specific. We choose not to include complementation strains in RNA sequencing experiments due to the large number of samples handling and associated costs.

Figure 3. These data are not rigorously controlled without genetic complementation, explain why some data in Figure 3 was generated at 24 hr and other data was generated at 48 hr, remove subbars in 3g. Please provide more clarification on Fig 3e-g because the normalization in these panels makes it appear as if there is little- or no-difference in the levels of 34S incorporation into the thiol metabolites.

The complementation strains used in all in vitro, ex vivo, and in vivo experiments showcase that the phenotypes associated with knockouts are gene-specific. We chose not to include complementation strains in Figure 3 experiments due to the large number of sample handling and associated costs.

The time points in the given experiment were chosen based on an initial pilot experiment. It is apparent that a longer duration is required to see the phenotypes associated with labelling compared to pool size. The differences observed are statistically significant.

Surfactant and SDS stress are used interchangeably in the text, legends, and figures. Please be consistent here.

We have modified the text in line with reviewer’s suggestion.

Consider re-wording the 1st paragraph on page 5 to better clarify how Trp, Lys, and His interact with the host immune cells.

We have modified the text in line with reviewer’s suggestion.

Cite the literature associated with the sulfur import system in Mtb on page 3 in the 2nd paragraph.

We have modified the text in line with reviewer’s suggestion.

The manuscript nicely describes the construction of a cysK2 mutant. It is unclear how the cysM mutant was generated. Please clarify, cite, or add the cysM mutant construction to this manuscript.

The details of CysM knockout and complementation strain generation has been previously published ([15]; Appendix Figure S4 & Methods). We have included the citation in the methods section of current manuscript.

Provide evidence that the small molecules used in Fig 6 are on target and inhibit the cysteine biosynthetic enzymes in whole bacteria. It is unclear how a MIC can be determined with these compounds in 7H9 ADC when deletion mutants grow just fine in this media. Is this because the compounds inhibit multiple cysteine synthesis enzymes and/or enzymatic targets in other pathways? To me, the data suggests that the compounds are hitting multiple enzymes in whole Mtb cells. Does cysteine supplementation reverse the inhibitory profiles with the compounds in Figure 6?

As mentioned in the text, all the compounds were ineffective in killing Mtb, likely because Lcysteine synthases are not essential during regular growth conditions. Hence, the MIC for cysteine inhibitors was very high - C1 (0.6 mg/ml), C2 (0.6 mg/ml), and C3 (0.15 mg/ml) opposed to the standard drug, isoniazid with MIC of 0.06 ug/ml. We agree with the reviewer that the experiments do not conclusively prove that these compounds specifically inhibit the cysteine synthases via "on-target inhibition" in Mtb cells. The inhibitors used in this study have been previously profiled in vitro [8]. However, one cannot rule out the hypothesis that these compounds might also have some off-target effects.